# A CASE FOR DATA VALUATION TRANSPARENCY VIA DVALCARDS

## ABSTRACT

Following the rise in popularity of data-centric machine learning (ML), various data valuation methods have been proposed to quantify the contribution of each datapoint to desired ML model performance metrics (e.g., accuracy). Beyond the technical applications of data valuation methods (e.g., data cleaning, data acquisition, etc.), it has been suggested that within the context of data markets, data buyers might utilize such methods to fairly compensate data owners. Here we demonstrate that data valuation metrics are inherently biased and unstable under simple algorithmic design choices, resulting in both technical and ethical implications. By analyzing 9 tabular classification datasets and 6 data valuation methods, we illustrate how (1) common and inexpensive data pre-processing techniques can drastically alter estimated data values; (2) subsampling via data valuation metrics may increase class imbalance; and (3) data valuation metrics may undervalue underrepresented group data. Consequently, we argue in favor of increased transparency associated with data valuation in-the-wild and introduce the novel Data Valuation Cards (DValCards) framework towards this aim. The proliferation of DValCards will reduce misuse of data valuation metrics, including in data pricing, and build trust in responsible ML systems.

## 1 INTRODUCTION

Recently, focus has shifted away from model-centric machine learning (ML) in favor of data-centric ML, whereby increased emphasis is placed on the importance of meaningful, high-quality data to a desired ML output Singh (2023). Within this paradigm, data valuation methods quantify the contribution of each datapoint (i.e. datum) to a given ML model performance metric (e.g., accuracy, loss, or a fairness measure such as equalized odds) Ghorbani & Zou (2019); Cook & Weisberg (1980); Arnaiz-Rodriguez & Oliver (2023a); Pang et al. (2024); Wang et al. (2024a). Increasingly, data valuation metrics as *influence functions* are utilized for various technical applications Hammoudeh & Lowd (2024); Sim et al. (2022); Fleckenstein et al. (2023), including data cleaning and subsampling Yoon et al. (2020); Ghorbani & Zou (2019); Koh & Liang (2017); Kwon & Zou (2021); Tang et al. (2021), data acquisition Ghorbani & Zou (2019); Kwon & Zou (2021); Jia et al. (2021), feature attribution Chen et al. (2023); Zhao et al. (2024), and active learning Ghorbani et al. (2022), with the specific application scenario influencing the choice of valuation function Sim et al. (2022). Additionally, data valuation techniques have been reappropriated to measure or modify the algorithmic fairness of ML systems Black & Fredrikson (2021); Arnaiz-Rodriguez & Oliver (2023a); Pang et al. (2024); Wang et al. (2024a). Within the context of data markets, it has been proposed that data buyers utilize data valuation methods for data pricing estimation in order to fairly compensate data owners according to their individual impact on model performance Laoutaris (2019); Paraschiv & Laoutaris (2019); Jia et al. (2019b;a). However, the practical limitations of in-the-wild data valuation are not yet well exposed.

Here, we highlight inherent properties of data valuation metrics - notably, bias and instability under simple algorithmic design choices - by examining diverse case studies. These experiments aim to address pragmatic questions: (1) Do standard data preprocessing techniques predictably alter data values?; (2) What are the technical side-effects of modifications to an ML system via data valuation? For instance: can data cleaning augment class imbalance?; and (3) What are the ethical side-effects of such modifications? Namely: are members of underrepresented groups more likely to yield undervalued data? Taken together, the context-dependent implications of these results underscore the

need for increased transparency regarding data valuation in-the-wild. Alternatively, the properties of data valuation metrics may limit their applicability to specific tasks entirely, as we argue in the case of data market pricing. Ultimately, we address the transparency gap by proposing a framework that we call DValCards, which accompany applications of data values and report the intended use, design choices, performance, and other critical information. We hope that the use of DValCards facilitates communication between creators, users, and affected parties of data valuation metrics, thereby encouraging appropriate use of the technology.

The code used in our experiments is available at: link.

## 1.1 RELATED WORK

**Data valuation.** The data valuation metrics we consider (leave one out (LOO), Truncated monte-carlo Shapley (TMC-Shapley), gradient Shapley (G-Shapley), etc.) were contextualized into an influence function taxonomy by Hammoudeh & Lowd (2024) and are introduced here in Section 2. Prior works have analyzed known limitations of data valuation methods with some proposing novel variants which attempt to address them. Zhou et al. (2023) find that Shapley estimators do not necessarily satisfy the fairness properties of true Shapley values. Schoch et al. (2022) develop a Shapley-based metric which better discriminates between in- and out-of-class contributions; here, we further analyze their method according to its impact on class imbalance. Ghorbani et al. (2020) propose a distributional Shapley framework to augment stability of data values under perturbations. Wang et al. (2024b) show that when applied to data selection, Data Shapley may perform no better than random selection without specific constraints on utility functions: for instance, when applied to homogeneous data. Wang & Jia (2023) discuss the instability of data value rankings across different model runs and propose a more robust data valuation metric; however, we demonstrate that their method (Banzhaf) still exhibits rank instability across algorithmic design choices. More generally, we focus specifically on LOO and Shapley-based values due to their popularity in real-world applications. Modeling choices have been found to result in varied feature attributions, with the specific task better informing the choice of Shapley-based approach Chen et al. (2020). More efficient Shapley value estimation methods have been proposed, e.g. Covert & Lee (2021); Chen et al. (2018); Kwon et al. (2021); Jethani et al. (2021). Yona et al. (2021) propose an extended Shapley method addressing joint credit assignment, and data valuation metrics have been extended to the federated learning setting, e.g. Wang et al. (2020); Liu et al. (2022); Song et al. (2019); Jiang et al. (2023).

**AI/ML transparency frameworks.** Modern ML transparency documentation frameworks are largely inspired by early documentation strategies including Data statements for natural language processing Bender & Friedman (2018), Datasheets for datasets Gebru et al. (2021), and Model cards for model reporting Mitchell et al. (2019). Existing frameworks are designed to enable users to comprehensively report essential characteristics of ML data, models, methods, or systems, and often cite similarities to nutrition labels or engineering datasheets Chmielinski et al. (2022); Krasin et al. (2017); Arnold et al. (2019). Frameworks may be contextualized for specific domains or applications, such as Healthsheets for healthcare applications Rostamzadeh et al. (2022), Reward reports for reinforcement learning Gilbert et al. (2023), or the Foundation Model Transparency Index Bommasani et al. (2023). Human-centric elements may be included for data reporting, such as the annotator demographic information recommended by Díaz et al. (2022). Data values are distinct from prior subjects of transparency documentation for a number of reasons, making existing frameworks inadequate for data value reporting; this is discussed in more detail in Section 4.

## 2 METHODOLOGY

**Experimental overview.** In this paper, we restrict our attention to the task of supervised classification. Let $\mathcal{D} = \{z_i = (\mathbf{x}_i, y_i)\}_{i=1}^n$ denote the training data, where $\mathbf{x}_i \in \mathcal{X} \subseteq \mathbb{R}^d$ are the features and $y_i \in \mathcal{Y}$ is the target class of the datum, $z_i$. Assume the model, $\mathcal{A}$, is trained on a subset of the data, $\mathcal{S} \subseteq \mathcal{D}$, to optimize the selected utility function, $\mathcal{V}(\mathcal{S}, \mathcal{A}) : 2^n \to \mathbb{R}$, where $2^n$ is the collection of all subsets of $\mathcal{D}$, including the empty set. To simplify notation, let $\mathcal{V}(\mathcal{S})$ denote $\mathcal{V}(\mathcal{S}, \mathcal{A})$. Throughout the paper, $\mathcal{V}(\mathcal{S})$ denotes the accuracy of the model on the validation (test) set, when trained on $\mathcal{S}$.

We utilize three diverse experiments as illustrative case studies, specifically:

1) **Metric instability:** 12 data imputation methods are applied as preprocessing techniques to 9 tabular datasets which are then used to train supervised classification models. The corresponding data values for each condition are reported using 4 data valuation metrics.

2) **Class imbalance:** 4 data valuation metrics are used to subsample data from 9 tabular datasets. The class imbalance is reported before and after subsampling using the balance estimates described in Appendix Section D.2.

3) **Underrepresented group bias:** 4 tabular datasets were analyzed to identify the prevalence of underrepresented attribute groups and their impact on 4 data valuation methods. Group and attribute representation is reported before and after subsampling using the balance estimates described in Appendix Section D.4.

**Datasets.** We selected 9 real-world, permissively licensed (CC BY), tabular classification datasets from the OpenML-CC18 benchmark license; Bischl et al. (2019). Dataset selection criteria are detailed in Appendix Section C.1. The datasets are reported by OpenML-CC18 labels: **18** (Mfeat-morphological), **23** (Contraceptive method choice), **31** (German credit), **37** (Pima Indians diabetes database), **54** (Vehicle silhouette), **1063** (KC2 Software defect prediction), **1068** (PC1 Software defect prediction), **1480** (Indian liver patient) and **40994** (climate-model-simulation-crashes). Basic dataset characteristics are listed in Appendix Table 1.

**Data preprocessing.** To test the impact of data imputation methods on data valuation metrics, we utilize tabular datasets with no missing values and induce missingness according to three percentages ($1\%, 10\%$ and $30\%$) and three patterns (missing completely at random (MCAR), missing at random (MAR) and missing not at random (MNAR)), as defined in Appendix Section A.1. Then we perform data imputation using 12 methods: row removal (i.e., discard all rows with any missing data values), column removal (i.e., remove attribute with missing data values), mean (i.e., replace a missing value with the mean of that attribute), mode (i.e., replace a missing value with the most frequent values within the attribute), $k$-nearest neighbor (KNN) Murti et al. (2019), optimal transport (OT) Muzellec et al. (2020), random sampling (i.e., randomly select samples from the attribute to fill the missing value), multivariate imputation by chained equations (MICE) van Buuren & Groothuis-Oudshoorn (2011), linear interpolation Huang (2021), linear round robin (LRR) Muzellec et al. (2020), MLP round robin (MLP RR) Muzellec et al. (2020), and random forest (RF) Hong & Lynn (2020). We include supplemental details in Appendix C.2.

**Data valuation.** The objective of the data valuation approach is to compute the datum value that reflects the marginal contribution of the datum to $\mathcal{V}$. Let the value of the datum $z_i$ to $\mathcal{V}$ be given by:

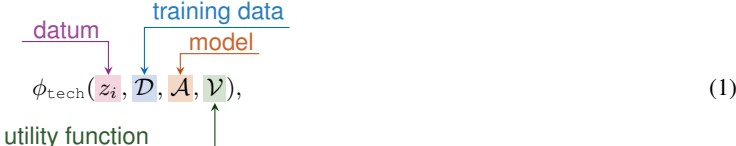

$$\phi_{\text{tech}}(z_i, \mathcal{D}, \mathcal{A}, \mathcal{V}), \tag{1}$$

where $tech$ is the datum valuation approach used to compute value of the datum. For simplicity, we use $\phi_{\text{tech}}(z_i)$ to denote $\phi_{\text{tech}}(z_i, \mathcal{D}, \mathcal{A}, \mathcal{V})$. In general, $\mathcal{V}(\mathcal{S}) - \mathcal{V}(\mathcal{S} \setminus \{z_i\})$ is defined as the marginal contribution of a datum to the utility function, $\mathcal{V}$. Different valuation approaches have different variants of this formulation.

For all experiments, we evaluate the data valuation approaches: truncated Monte Carlo Shapley (TMC-Shapley) Ghorbani & Zou (2019), gradient Shapley (G-Shapley) Ghorbani & Zou (2019), and leave one out (LOO) Cook & Weisberg (1980). See Appendix Sections A.2 for method descriptions and C.3 for learning algorithm and additional details. Additionally, we analyze Banzhaf Wang & Jia (2023) with respect to metric instability, class-wise Shapley (CS-Shapley) Schoch et al. (2022) for class imbalance analysis, and FairShap Arnaiz-Rodriguez & Oliver (2023a) fairness-based metrics exclusively for the fairness experiment, beyond the standard metrics.

# 3  RESULTS

## 3.1  METRIC INSTABILITY CASE STUDY: DATA IMPUTATION

We find that varying the applied data imputation method results in appreciable variation of data values, with all other experimental conditions held constant (see Figure 1). Notably, the data rank order change is statistically significant when cross-comparing data values corresponding to any two differing imputation methods, according to Kendall's $\tau$ coefficient: $\tau < 1$ and $p < 0.05$ Kendall (1938). This trend holds across all the data valuation methods considered (TMC-Shapley, G-Shapley, LOO and Banzhaf); see Appendix Figure 5.

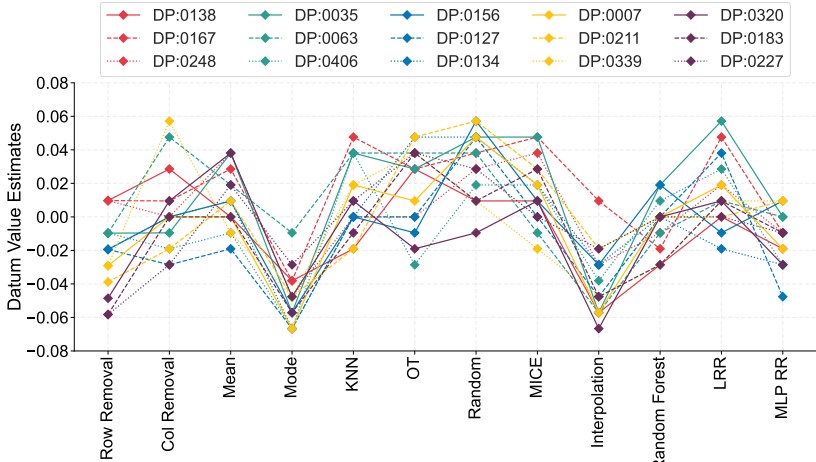

(a) Variation in data values for fixed data points by imputation method

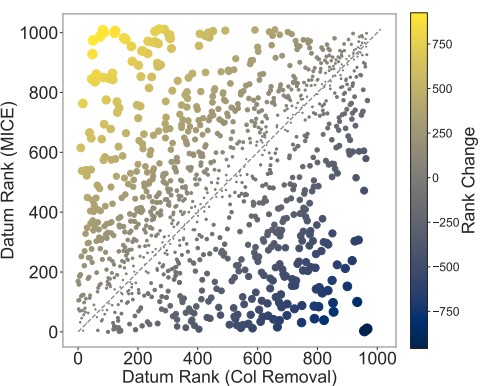

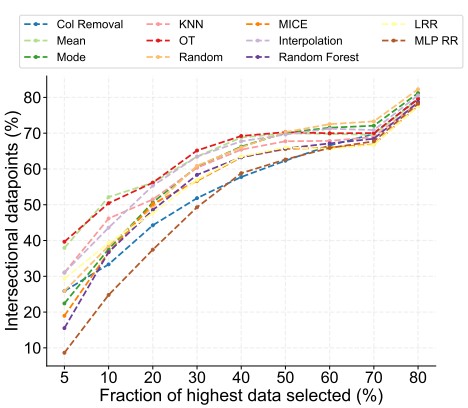

(b) Cross-comparison of datum rank values by imputation method

(c) Variations in value-selected data subsets by imputation method

Figure 1: Data values are unstable to choice of data preprocessing method. (**a**) Leave-one-out (LOO) value estimates vary as a function of imputation method; data points are selected to span 5 quintiles of data value scores from the row removal results (grouped by color). By cross-comparing value estimates by imputation method, it is clear that value rank order varies in addition to raw values. (**b**) TMC-Shapley value-based data ranks are compared across two different imputation methods (column removal and MICE) to assess agreement. The Kendall $\tau = 0.3214$, $p$-value $< 0.05$, indicating a statistically significant positive correlation but not agreement, as can be observed by the scattering of points away from the diagonal (grey dashed line). Point size indicates scale of rank change; see also Appendix Figure 7 for changes in value and rank across all points. (**c**) The percentage of shared points between high data value sets as a function of various imputation methods and row removal as the baseline method. Analogous plots including low data value selection are provided in Appendix Figure 9.

**Variance in data values.** Figure 1a shows a snapshot of 15 fixed data points in dataset 1063 (KC2 Software defect prediction, with MNAR-1) and the variance of leave-one-out (LOO) data values after various imputation methods were performed. The points were selected as non-imputed values spanning the quartiles of the initial data value set condition (MNAR, row removal), with three points per bin. This snapshot demonstrates a crucial point: that data values can vary significantly according to imputation method applied; not only in an absolute sense (which may be meaningless to compare cross-system), but even causing drastic relative changes in score for non-imputed data points. For completeness, mean data values are reported systematically across all imputation methods and data valuation metrics considered in Appendix Figure 6. In general, utilizing data imputation methods tends to increase the average data value (see Appendix Table 2a and Figure 8) and, in some cases, the maximum data value (see Appendix Table 2b).

**Variance in data rank.** To illustrate datum rank changes across all data points for a single dataset, we select two imputation methods (column removal and MICE) and cross-compare how datum rank is impacted for all data values in dataset 23 (Contraceptive method choice, with TMC and MAR-30). These results are shown in Figure 1b; we would expect a stable valuation metric to reasonably maintain consistent rank scores, and display a trend along the diagonal (shown as the grey dotted line). The wide variability of rank scores in this case study suggests that data value instability may not uniquely impact high or low data values. To better systematically assess rank order changes, we report the Kendall's $\tau$ coefficient across each pair of imputation methods acting on dataset 37 (Pima Indians Diabetes Database, with MNAR-10) for all data valuation metrics considered in Figure 5. We find that the imputation method of row removal and the data valuation metric LOO are associated with significant rank changes in comparative analyses across imputation strategy.

**Implications to data subsampling.** Indeed, for many practical applications of data valuation metrics, the data values are used to select a subset of the initial dataset according to highest or lowest data values, such as in data cleaning. Thus, we ask: are data valuation metrics capturing the same points as the sub-selected data fraction varies, if only the imputation method is modified? We show that the same points are not necessarily captured in Figure 1c; in this, we present the percentage of data points captured by TMC values following applications of different imputation methods when compared to the baseline method, row removal (dataset 23, MCAR-30). Analogous plots with both the highest- and lowest-valued data fractions are shown for each of the metrics considered in Appendix Figure 9. Moreover, data values assessed prior to imputation could lead to the premature disposal of otherwise high-valued data as assessed post-imputation. In the following section, we explore class-based implications of value-based data subsampling.

## 3.2 TECHNICAL IMPACT CASE STUDY: CLASS IMBALANCE

We find that the distribution of data values can vary greatly as a function of class membership and data valuation metric. As a result, data value-based subsampling may increase class imbalance.

**Data value distributions may be class-dependent.** We observe that most standard data valuation metrics exhibit class-based bias, with sample results shown in Figures 2a-2b for TMC-Shapley and G-Shapley. All results in Figure 2 are shown for dataset 40994 (climate-model-simulation-crashes) under MCAR-10 and random imputation. Notably, the associated binary classes are imbalanced, with the larger class ("simulation success") comprising 91.3% of the data. In Figures 2a-2b, the Shapley-based data valuation metrics can be seen to produce lower data values for the less frequent class ("simulation failure", blue) than the more frequent class ("simulation success", orange). By contrast, the application of the class-wise Shapley (CS-Shapley) metric reduces the class-based bias on the same data: see Figure 2c, in which the distinct classes correspond to similar data value distributions. This trend is unsurprising, as CS-Shapley was developed to better discriminate between training instances' in-class and out-of-class contributions to a classifier. However, the differences observed across data valuation metrics indicate the utility of clear transparency documentation, especially given the impact of the choice of data valuation method on other performance metrics. Additional class-based value distribution plots are shown in Appendix Figure 10 for diverse datasets and metrics.

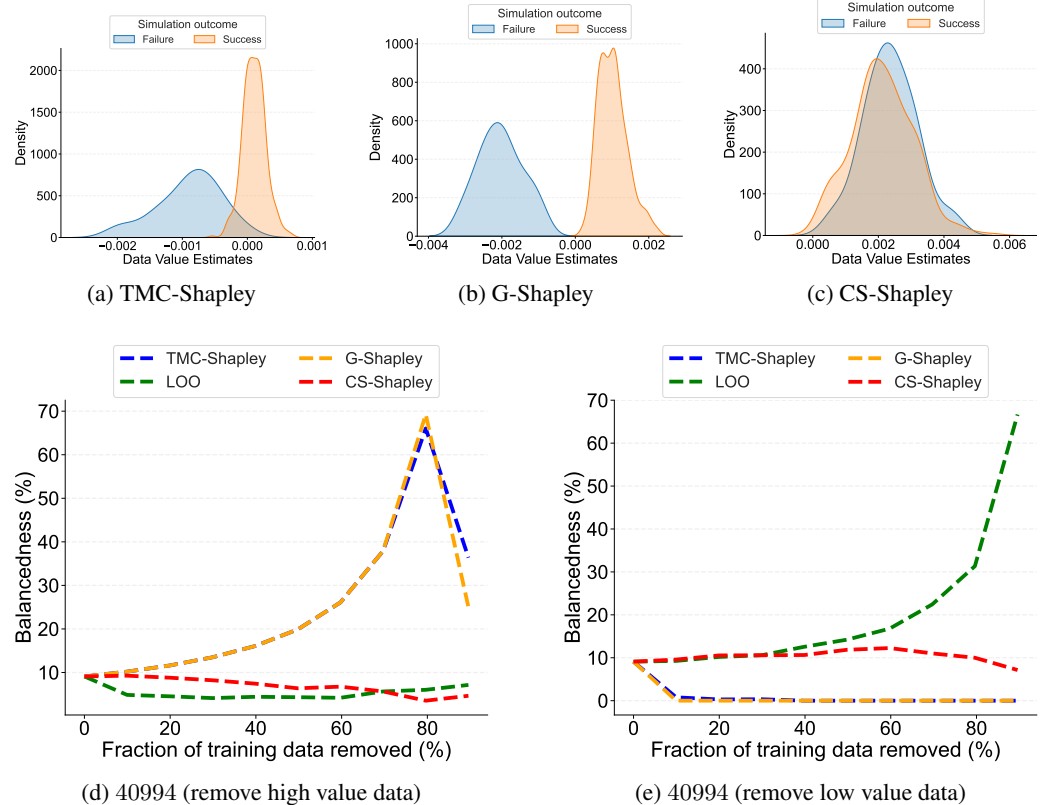

Figure 2: Data values and class imbalance. (**a**, **b**, and **c**) Data value distributions according to three valuation metrics (TMC-Shapley, G-Shapley, and CS-Shapley, respectively), for a binary classifier with 91.3% simulation-outcome *success* (dataset 40994, MCAR-10). We observe marked class-based differences in data value distributions for TMC-Shapley and G-Shapley; by contrast, class-wise Shapley (CS-Shapley) improves consistency between classes. (**d**, **e**) Class balance (as defined in Appendix D as $b$) versus percentage of data removed, as a function of four data valuation metrics (TMC-Shapley, G-Shapley, LOO and CS-Shapley). Value-based data subsampling may impact class imbalance. In this example, TMC-Shapley and G-Shapley increase class balance with removal of high-value data and decrease balance with removal of low-valued data.

**Value-based subsampling may impact class balance.** To illustrate how class imbalance can change as a function of value-based data subsampling, we show class balancedness as a function of percentage removed data for each metric, e.g. in Figures 2d-2e. Given the same initial dataset with imbalanced classes, we observe that TMC-Shapley and G-Shapley result in reduced class balance as low-valued data is removed; this is indicative of data removal from the lower-valued, smaller class ("simulation failure") corresponding to the value distributions shown in Figures 2a-2b. The opposite trend holds as high-valued data is removed, indicative of data pulled from the majority class, until an inflection point is reached. By contrast, CS-Shapley results in a relatively consistent class balance when either high- or low-valued data is removed. LOO results in reduced class balance as high-valued data is removed and increased class balance as low-valued data is removed. Analogous plots for diverse datasets and imputation methods may be found in Appendix Figure 11. We systematically review all datasets, imputation methods, missingness conditions and value metrics according to their impact on class balance following subsampling in Appendix Table 3 and Table 4, corresponding to removal of low- or high-valued data, respectively. Results are reported according to absolute class balance scores (i.e., balancedness ¡ 0.25) and to relative class balance with respect to the original dataset. We find that when 20% of low-valued data is removed, the absolute and relative class balance worsens for most datasets; the removal of high-valued data does not generally reduce class balance. Finally, the choice of metric may have diverse effects on downstream performance metrics, such as accuracy (see Appendix Figure 20) or attribute balance (see Section 3.3).

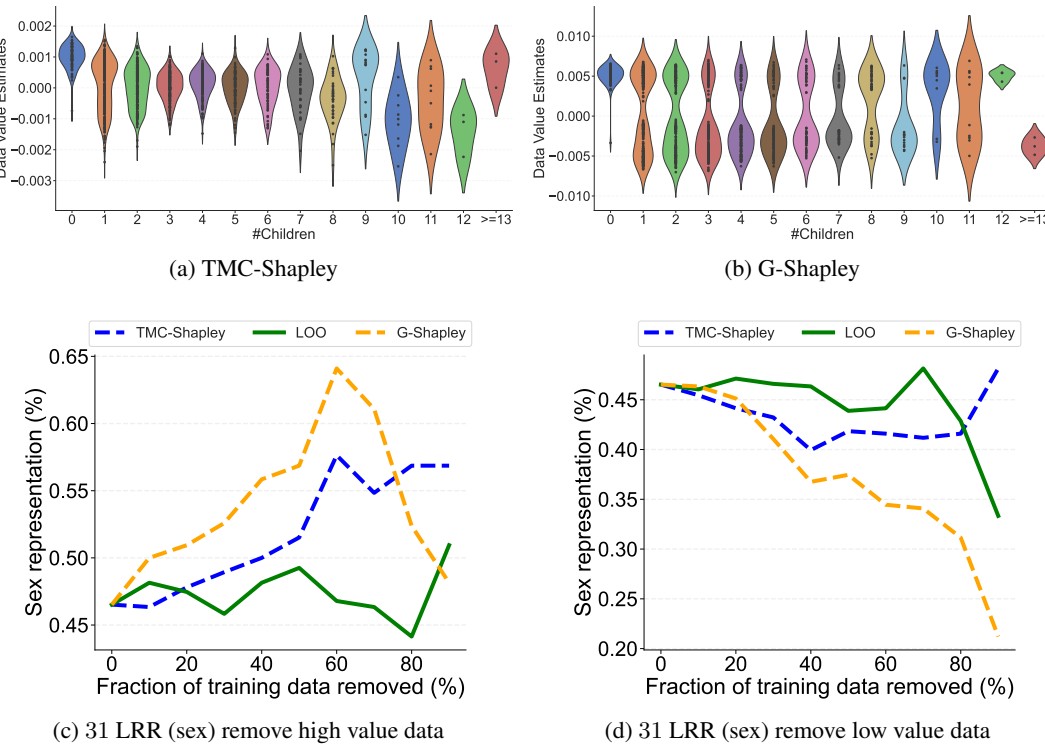

(a) TMC-Shapley

(b) G-Shapley

(c) 31 LRR (sex) remove high value data

(d) 31 LRR (sex) remove low value data

Figure 3: Data values and attribute group. (**a**, **b**) Data value distributions according to two valuation metrics (TMC-Shapley and G-Shapley, respectively), by attribute group "number of children ever born" (dataset 23, MAR-10). We observe marked attribute-based differences in data value distributions with variance across valuation metric. In (**a**), removal of low-valued data may disproportionately remove data from underrepresented attribute groups (i.e, greater "number of children ever born"). (**c**, **d**) Percentage binary sex representation (as defined in Appendix D as *g*) versus percentage of data removed, as a function of three data valuation metrics (TMC-Shapley, LOO, and G-Shapley). Value-based data subsampling may impact attribute imbalance. In this example, TMC-Shapley and G-Shapley tend to increase percentage sex representation with removal of high-value data and decrease representation with removal of low-valued data.

### 3.3 ETHICAL IMPACT CASE STUDY: POTENTIAL UNDERVALUATION OF MARGINALIZED GROUPS

We find that data valuation metrics may exhibit attribute-based bias as a function of dataset and preprocessing conditions. As a result, the choice of metric in the context of specific downstream applications, like data subsampling, may impact attribute balance in an unpredictable manner. When an attribute (e.g. skin tone, gender) denotes a sensitive characteristic associated with protected or marginalized groups of people, the potential for selective removal of these infrequent data samples is ethically (and possibly legally) problematic. Similar implications apply to other value-based applications including pricing in data markets.

**Data values by attribute group.** Our experiments show that data valuation metrics manifest distinct and potentially biased distributions across attribute groups. Two examples are provided in Figures 3a - 3b, which display the variance in TMC-Shapley and G-Shapley values according to attribute for dataset 23 (Contraceptive method choice, with MAR-10); here, the attribute of interest is "number of children ever born". In these, we observe distinct distributions for data value across the various attribute classes. Lower data values in TMC-Shapley were often associated with underrepresented groups (Figure 3a), i.e. with greater "number of children ever born"; thus downstream subsampling based on low-value data removal may pull relatively more data from these underrepresented groups. Heterogeneity of distributions according to attribute group may be observed under a multitude of experimental conditions: additional analogous plots to 3a - 3b are shown in Appendix

Figure 13. These show that CS-Shapley may also produce distinct distribution clusters for specific attributes, and thus a method chosen to protect class balance may still result in the selective removal of data from underrepresented attribute groups, or other issues caused by data undervaluation.

**Subsampling and attribute balance.** Sample plots in Figures 3c - 3d illustrate how attribute balance may be impacted by value-based subsampling. For dataset 31 (German credit, LRR, MCAR-30), we see that as an increasing fraction of low-valued data is removed, TMC- and G-Shapley tend to result in worsening female-to-male binary sex representation, with generally smaller effects resulting from LOO. Analogous plots to Figures 3c - 3d for diverse imputation methods can be found in Appendix Figure 15, and imputation method is systematically assessed for its impact on attribute balance and equalized-odds difference (EOD, a fairness metric) for age and sex in Appendix Figure 16 and Figure 17, respectively. We cross-compare model accuracy with EOD for both binary sex and age in Appendix Figure 20 for dataset 31 (German credit, mean, MAR-30), as increasing fractions of data are removed, demonstrating that EOD is not necessarily correlated with changes in predictive accuracy.

For all missingness conditions, imputation methods and standard value metrics we present results in which subsampling improves EOD fairness for sex (see Appendix Table 5) and age (see Appendix Table 6) on dataset 31 (German credit, mean, MAR-30). From this systematic analysis we find that across all conditions, subsampling typically does not improve EOD fairness. Similarly, we assess the impact on attribute representation balance for all conditions, for sex (see Appendix Table 7) and age (see Appendix Table 8). The results are found to vary more widely for attribute representation balance, and this may be impacted by the initial attribute representation balance from the original dataset.

For comparison, we additionally show the distribution of data values by attribute group and class according to accuracy and three fairness metrics (equalized odds "Odds", average absolute equalized odds "Odds2", and equal opportunity "EOp") using the protocol described in Arnaiz-Rodriguez & Oliver (2023b) on select datasets (see Appendix Figure 14 and Equation (7)). As expected, the distribution of accuracy- and fairness-based values display distinct characteristics, as the removal of points of low influence to accuracy may negatively impact fairness outcomes; this is assessed systematically in Appendix Figures 18d and 19 across binary sex and age.

### 3.4 FAIR COMPENSATION

We briefly comment on the oft-cited recommendation that data valuation metrics be utilized as, or a major constituent of, a data pricing scheme. Our results indicate that a naïve utilization of the LOO and Shapley-based metrics is unsuitable for establishing equitable compensation. In Section 3.1, we illustrate the instability of LOO, TMC-Shapley and G-Shapley to 12 common data preprocessing (imputation) methods. Such instability induces no confidence in data metrics as a pricing scheme; that is, it is unclear to data market participants how minor algorithmic design choices may impact data costs. Likewise, control over algorithmic design may provide data buyers with a mechanism by which to artificially adjust data prices to the detriment of data owners. In Section 3.3, we demonstrate the potential for attribute group bias in data values; as a data pricing scheme, this puts data buyers at risk of explicitly undervaluing data offered by members of marginalized groups or other "outlier" types. (Interestingly, such an effect could make homogeneous data more expensive from a buy-side perspective.) Notably, data valuation metrics are *unfair* by design, as evidenced by their utility for data outlier removal and cleaning. Furthermore, we argue that data valuation metrics lack properties of an effective economic pricing strategy: for instance, an inherent asymmetry is given to the seller, as data owners must submit their data in order to receive an assigned price. Prior works have highlighted this and a number of other practical challenges with the use of data valuation metrics as a pricing scheme, which include computational expense (Hammoudeh & Lowd (2024)), the handling of replicated data Xu et al. (2021); Agarwal et al. (2019); Wang & Jia (2023); Ohrimenko et al. (2019), the translation to a monetary value Coyle & Manley (2023), asymmetry in data marketplace design Azcoitia & Laoutaris (2022); Agarwal et al. (2019); Han et al. (2023), privacy leakage Tian et al. (2022); Wang et al. (2023); Kang et al. (2024) and protections against strategic sellers Castro Fernandez (2022); Agarwal et al. (2019). In many practical contexts, fair and consistent compensation may more readily be obtained by assigning data values *a priori* and decoupling values from learning algorithms and performance metrics.

## 4 DVALCARDS FOR DATA VALUATION TRANSPARENCY

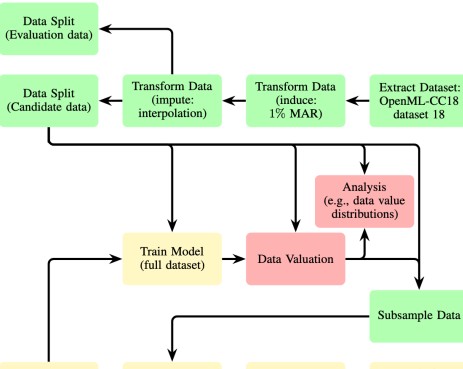

**DValCard - Multiple Features Dataset: Morphological**

This DValCard is developed for demonstration purposes. It was developed by the authors of this paper in May 2024. Results from dataset **18** (Mfeat-morphological) are utilized.

**System Flowchart**

- **DVal in the life cycle context**. Data valuation is conducted during the data preprocessing stage of model training (see Figure 9). The DVal candidate data is a subset of, or equal to, the model training data.

**DVal Candidate Data**

- **Data information**. The multiple features morphological dataset (Duin, 1998) contains six morphological features that describe handwritten numerals $(0-9)$ taken from a set of Dutch utility maps. It includes approximately $10\%$ of each digit, totaling 1600 instances. Although the original dataset has no missing values, we introduced $1\%$ missingness via a random missingness pattern for experimentation.
- **Data preprocessing**. Interpolation imputation method (Huang, 2021) is used to impute missing data.

**DVal Method**

- **DVal technique**. G-Shapley (Ghorbani & Zou, 2019).
- **Learning algorithm**. Logistic regression model, with the linear *solver* and *max_iter* 5000 for data valuation and classification.
- **Performance metric**. Learning algorithm accuracy score: the sum of true positives and true negatives out of all algorithmic predictions.
- **Evaluation data**. The evaluation data consists of 400 instances with 6 morphological features describing handwritten numerals $(0-9)$ extracted from a collection of Dutch utility maps. The dataset was the $20\%$ split from the $80/20$ split Multiple Features Dataset: Morphological (Duin, 1998) part of which was used for data valuation.

**DVal Report**

- **Data values**. The maximum data value is $0.0178875$ and the minimum data value is $-0.0156906$. The distribution of data values shows an over-representation of the numeric digits 3, 4, 5, and 6 in the discarded data (refer to Figure Figures 4 and 7).
- **Chosen/included instances**. $80\%$ of instances with the highest data values were included (see Figure 5).

**Ethical Statement and Recommendations**

- **Intended users, and in/out-of-scope use cases**. This card is demonstrative and meant for researchers, engineers and all those interested in data valuation.
- **Potential ethical issues to consider**. The chosen data valuation scheme does not perform well on the intended task, as seen in Figure 6. We caution against using it for data subsampling.
- **Legal considerations**. The DValCard and experimentation details are provided under the MIT license (license, a).
- **Environmental considerations**. A CPU worker is used for compute, with full hardware specs provided in Appendix C.3.
- **Recommendations**. Due to accuracy decreases resulting from application of value-based subsampling, we recommend the consideration of alternative data valuation methods, e.g. CS-Shapley to better handle imbalanced classes (Schoch et al., 2022).

Figure 4: Data values distribution

Figure 6: Low value data removal

Figure 5: The distribution of numeric digits for **included** instances

Figure 7: The distribution of numeric digits for **excluded** instances

(a) DValCard Example

(b) System Flowchart

Figure 4: Illustration of the DValCard example (4a), and a system flowchart that contextualizes the use of data valuation for increased transparency in the DValCard (4b). Processes most closely associated with data in green; data valuation in red; and the ML model in yellow. Evaluation data was used in both analysis processes (omitted arrows indicating this for clarity) to assess the contribution of DVal Candidate data to the trained model and to evaluate ML models trained on subsampled datasets.

Given the limitations of data valuation metrics explored in previous sections, we propose a transparency framework to promote confidence in, and appropriate use of, such metrics. There exist key differences between data valuation methods and the subjects of existing transparency documents: in particular, data values can (1) form part of the *data life cycle*; (2) form part of the *model life cycle*; or (3) be utilized as standalone measures. Within a data life cycle, data values may be used for dataset curation, e.g. in explanations of data diversity, density or association (Mitchell et al., 2023) or instance removal (Gebru et al., 2021). Within a model or system life cycle, data values are used for model training, e.g. for data weighting, selection, cleaning and preprocessing Arnaiz-Rodriguez & Oliver (2023b); Yoon et al. (2020); Koh & Liang (2017); Kwon & Zou (2021); Tang et al. (2021); Ghorbani & Zou (2019); Kwon & Zou (2021). Furthermore, data values may be independently used for tasks including data pricing. Consequently, existing transparency documents do not well capture the flexibility required for data valuation reporting: system cards Alsallakh et al. (2022) assume the existence of ML models contained within a broader pipeline, while datasheets Gebru et al. (2021) exclude models entirely, as examples. Another key feature of data values is that accurate reporting of *when* values are computed is essential, with respect to other ML system components; in Section 3.1 we illustrate the impact of simple preprocessing choices on data values. This motivates our recommendation that DValCard authors include ML system flowcharts to clearly detail the order of operations. Correspondingly, certain performance measures, such as attribute balance, may change as the result of data value-based processes, such as value-based subsampling (see Section 3.3). Thus, we encourage reporting performance before and after the data value application. Figure 4 illustrates an example DValCard, with the main sections highlighted in blue, and Appendix H includes details of the proposed general sections of the DValCard, intended to flexibly integrate the "ingredients" of data valuation methods and better elucidate system performance in the context of intended use.

## 5 CONCLUSIONS

We introduce the DValCards framework to support decision-making and promote the appropriate use of data valuation methods. Through three case studies, we demonstrate notable disparities of data valuation in practice: the variability in data values caused by common data preprocessing techniques (Section 3.1), the influence of data values on class imbalances (Section 3.2), and the disparate valuation of underrepresented attribute groups (Section 3.3). We argue that comprehensive and transparent documentation—covering appropriate data valuation methods use, implementation specifics, performance metrics, and fairness considerations—will significantly improve usage.

**Limitations** Our experiments primarily centered on a small set of data valuation metrics: TMC-Shapley, G-Shapley, and LOO. We selected these methods based on three criteria: they are the most frequently cited in the literature, serve as a foundation for many modern methods that often refine or address the limitations of these fundamental approaches (e.g., CS-Shapley), and are widely applied in data pricing and data markets, with Shapley values being particularly prominent. While alternative metrics may exist that better address some of the technical and ethical challenges we examine, transparency remains essential to foster clear communication between stakeholders in practice.

Moreover, our choice to highlight practical case studies is inherently restrictive; for example, we do not extend beyond the tabular supervised classification domain nor explore preprocessing methods beyond imputation. Additionally, the OpenML-CC18 benchmarking datasets we utilize do not have comprehensive associated transparency documentation (e.g., datasheets). Thus, in some settings, the exact provenance of the original data and the use of ethical curation practices remain unclear. To the best of our knowledge, we are the first to empirically study the practical limitations of data valuation in real-world use cases and propose a specific framework for data valuation transparency. We hope that future researchers can test the framework in practical applications.

Lastly, challenges may arise in enforcing the DValCards standard and incentivizing researchers and practitioners to adopt and implement the documentation effectively. The current proposed DValCard template aims to initiate a discussion and encourage practitioners and researchers to modify it to ensure accurate and comprehensive documentation of the data valuation process. With agreement on the standard, practitioners and researchers can integrate the DValCard into their documentation. We believe we can successfully follow a similar route taken by other documentation and transparency methods to incentivize researchers and practitioners to incorporate DvalCards into existing documentation frameworks.

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

# A    PROBLEM DEFINITIONS

## A.1    DATA MISSINGNESS DEFINITIONS.

Assume data is partitioned into observed and missing data: $\mathcal{X}_{init} = (\mathcal{X}_{init}^{obs}, \mathcal{X}_{init}^{miss})$. Let $\mathbf{R} \in \{0,1\}^{n \times d}$ be a random variable that denotes a missingness pattern where $\mathbf{R}_{ij} = 1$ if $\mathcal{X}_{init,ij}$ is observed and $0$ otherwise. The probability distribution of $\mathbf{R}$ is denoted by $\mathbb{P}_{\mathbf{R}}$ and parameterized by $\epsilon$. Then, the probability of the missingness patterns are given below.

- With MCAR, the missingness is independent of the variables and observations. The probability of MCAR is defined as $\mathbb{P}_{\mathbf{R}}(\mathbf{R}|\epsilon)$.

- The likelihood of a missing value in MAR is dependant on only the observable data. The probability for MAR can be defined as $\mathbb{P}_{\mathbf{R}}(\mathbf{R}|\mathcal{X}_{init}^{obs}, \epsilon)$.

- MNAR is when missing data is neither MCAR nor MAR. The missing data depends equally on the missing and observed data. The MNAR probability is defined as $\mathbb{P}_{\mathbf{R}}(\mathbf{R}|\mathcal{X}_{init}^{obs}, \mathcal{X}_{init}^{miss}, \epsilon)$.

## A.2    DATA VALUATION METRIC DEFINITIONS.

**Leave One Out (LOO)**    is an algorithm which is more commonly used in model training for cross validation or model selection. In this setting, the model is trained on $n - k$ data points and then evaluated and fine-tuned on $k$ (here, $k = 1$) data points. Similarly, to compute the LOO value of a datum, the datum is excluded from the training dataset Cook & Weisberg (1980). Valuation of a datum with LOO is computed as:

$$\phi_{loo}(z_i) = \mathcal{V}(\mathcal{S}) - \mathcal{V}(\mathcal{S} \setminus \{z_i\}), \tag{2}$$

where here, the training subset $S$ is the entire training dataset, $\mathcal{D}$.

**The Shapley value**    is a solution concept in cooperative game theory for semi-values. Due to several of its axiomatic properties, Ghorbani & Zou (2019) suggested the use of Shapley values to compute the value of a datum to machine learning. The Shapley value of a datum is defined as

$$\phi_{\texttt{shapley}}(z_i) = \frac{1}{n} \sum_{\mathcal{S} \subseteq \mathcal{D} \setminus \{z_i\}} \frac{1}{\binom{n-1}{|\mathcal{S}|}} \big[ \mathcal{V}(\mathcal{S} \cup \{z_i\}) - \mathcal{V}(\mathcal{S}) \big]. \tag{3}$$

The axiomatic properties of Shapley values that make it favourable for data valuation include the following:

1) Null player: If for all $\mathcal{S} \subseteq \mathcal{D}, \mathcal{V}(\mathcal{S}) = \mathcal{V}(\mathcal{S} \cup \{z_i\})$, then $\phi_{\texttt{shapley}}(z_i) = 0$.

2) Efficiency: $\sum\limits_{z_i \in \mathcal{D}} \phi_{\texttt{shapley}}(z_i) = \mathcal{V}(\mathcal{D})$.

3) Symmetry: If $i$ and $j$ are such that $\mathcal{V}(\mathcal{S} \cup \{z_i\}) = \mathcal{V}(\mathcal{S} \cup \{z_j\})$, then $\phi_{\texttt{shapley}}(z_i) = \phi_{\texttt{shapley}}(z_j)$.

4) Linearity: For any 2 utility functions $\mathcal{V}_1$ and $\mathcal{V}_2$, and $\alpha_1, \alpha_2 \in \mathbb{R}$,
   $\phi_{\texttt{shapley}}(z_i, \alpha_1 \mathcal{V}_1 + \alpha_2 \mathcal{V}_2) = \alpha_1 \phi_{\texttt{shapley}}(z_i, \mathcal{V}_1) + \alpha_2 \phi_{\texttt{shapley}}(z_i, \mathcal{V}_2)$.
   Additionally, $\phi_{\texttt{shapley}}(z_i, \mathcal{V}_1 + \mathcal{V}_2) = \phi_{\texttt{shapley}}(z_i, \mathcal{V}_1) + \phi_{\texttt{shapley}}(z_i, \mathcal{V}_2)$.

Despite these properties, the true Shapley value is computationally complex; it is exponential in the number of data points. TMC-Shapley and G-Shapley are approximations of the Shapley value designed to counter this complexity, among others Ghorbani & Zou (2019); Jia et al. (2019b; 2020).

**TMC-Shapley**    was proposed by Ghorbani & Zou (2019) as a truncated Monte Carlo approximation of the Shapley value. In this, a scan through sampled permutations is performed to compute truncated marginal contributions to $\mathcal{V}(\mathcal{S})$ within a performance tolerance of $\mathcal{V}(\mathcal{D})$ and assign 0

marginal contribution to other data points within the permutation. If there are $n!$ permutations of data points and $\Pi$ is the uniform distribution over all of them, and $\mathcal{S}_\pi^i$ is the set of data points coming before datum $z_i$ in permutation $\pi \in \Pi$, then:

$$\phi_{\texttt{tmc-shapley}}(z_i) = \mathbb{E}_{\pi \sim \Pi}\big[\mathcal{V}(\mathcal{S}_\pi^i \cup \{z_i\}) - \mathcal{V}(\mathcal{S}_\pi^i)\big]. \tag{4}$$

**G-Shapley** was proposed in the same work as a related, gradient-based Monte Carlo approximation of Shapley. The algorithm approximates the marginal contribution of the datum $z_i$ by taking gradient descent step using $z_i$ and computing the difference in $\mathcal{V}$. We refer the reader to Ghorbani & Zou (2019) for a more detailed description of the algorithms.

**CS-Shapley** is a Shapley value estimation variant that differentiates between the contribution of a $z_i$ to its own class and to other classes (Schoch et al., 2022). We refer the reader to Schoch et al. (2022) for a detailed description of the method.

$$\phi_{\texttt{cs-shapley}}(z_i) = \frac{1}{2^{|\mathcal{D}_{-yi}|}} \sum_{\mathcal{S}_{y_i} \subseteq \mathcal{D}_{y_i} \setminus \{z_i\}} \frac{1}{\binom{n-1}{|\mathcal{S}_{y_i}|}} \big[\mathcal{V}_{y_i}(\mathcal{S}_{y_i} \cup \{z_i\}|\mathcal{S}_{-y_i}) - \mathcal{V}_{y_i}(\mathcal{S}_{y_i}|\mathcal{S}_{-y_i})\big]. \tag{5}$$

**Banzhaf** as proposed by Wang & Jia (2023), is a semivalue-based data valuation scheme with increased robustness across model runs compared to TMC-Shapley. We refer the reader to Wang & Jia (2023) for a detailed description of the method.

$$\phi_{\texttt{banzhaf}}(z_i) = \frac{1}{2^{|\mathcal{D}|-1}} \sum_{\mathcal{S} \subseteq \mathcal{D} \setminus \{z_i\}} \big[\mathcal{V}(\mathcal{S} \cup \{z_i\}) - \mathcal{V}(\mathcal{S})\big]. \tag{6}$$

**FairShap** as proposed by Arnaiz-Rodriguez & Oliver (2023b) is a variant of Shapley value estimation building on the work of Jia et al. (2019a) to compute the marginal contribution of $z_i$ by means of $k$-NN approximation and the validation dataset $\mathcal{T}$. FairShap considers the family of data valuation methods centering error rate fairness metrics. With $\Phi_{i,j}$ defined as the marginal contribution of $z_i$ to the probability of correct classification of the test point $\mathbf{x}_j \in \mathcal{T}$, the definition of $\texttt{fairshap-SVAcc}(z_i)$ is:

$$\phi_{\texttt{fairshap-SVAcc}}(z_i) = \frac{1}{m} \sum_{j=1}^{m} \Phi_{i,j}. \tag{7}$$

We refer the reader to Arnaiz-Rodriguez & Oliver (2023b) for thorough details on computation of the $\phi_{\texttt{fairshap-SVAcc}}(z_i)$'s fairness derivative data values: $\phi_{\texttt{fairshap-SVEOp}}(z_i)$ (marginal contribution of $z_i$ to equal opportunity), $\phi_{\texttt{fairshap-SVOdds}}(z_i)$ (marginal contribution of $z_i$ to average equalized odds), and $\phi_{\texttt{fairshap-SVOdds2}}(z_i)$ (marginal contribution of $z_i$ to average absolute equalized odds).

# B  DATASET CHARACTERISTICS

| Name (ID) | Source | #Classes | #Features | Train | Test |
|---|---|---|---|---|---|
| Mfeat-morphological (**18**) | https://www.openml.org/search?type=data&sort=runs&status=active&id=18 | 10 | 7 | 1600 | 400 |
| Contraceptive method choice (**23**) | https://www.openml.org/search?type=data&status=active&id=23 | 3 | 10 | 1178 | 295 |
| German credit (**31**) | https://www.openml.org/search?type=data&status=active&id=31 | 2 | 20 | 800 | 200 |
| Pima Indians diabetes database (**37**) | https://www.openml.org/search?type=data&status=any&id=37 | 2 | 7 | 614 | 154 |
| Vehicle silhouette (**54**) | https://www.openml.org/search?type=data&status=any&id=54 | 4 | 18 | 676 | 170 |
| KC2 Software defect prediction (**1063**) | https://www.openml.org/search?type=data&status=active&id=1063 | 2 | 22 | 417 | 105 |
| PC1 software defect prediction (**1068**) | https://www.openml.org/search?type=data&status=active&sort=runs&id=1068 | 2 | 22 | 887 | 222 |
| Indian liver patient (**1480**) | https://www.openml.org/search?type=data&status=any&sort=runs&id=1480 | 2 | 10 | 466 | 117 |
| climate-model-simulation-crashes (**40994**) | https://www.openml.org/search?type=data&status=any&sort=runs&id=40994 | 2 | 21 | 432 | 108 |

Table 1: Basic characteristics of the OpenML-CC18 tabular datasets used in the experiments.

# C  METHODOLOGY: SUPPLEMENTAL DETAILS

## C.1  DATASET SELECTION CRITERIA

9 datasets were sub-selected from 69 OpenML-CC18 datasets according to the following criteria: (1) the data contains no missing values; (2) the existence of at most 10 classes; and (3) the existence of a number of data features within the range $(5, 25]$.

## C.2  DATA PREPROCESSING

**Missingness.** Missingness patterns were selected among three patterns defined by Rubin (1976), in which data is: missing completely at random (MCAR), missing at random (MAR) and missing not at random (MNAR). We define them here in section A.1. To vary data missingness, we first select one or more fixed features: specifically, feature 3 for datasets with $\leq 8$ features, and features 2 and 7 for datasets with $\geq 9$ features. For each missingness pattern (MCAR, MAR, and MNAR), data missingness is induced according to three percentages: $1\%, 10\%$ and $30\%$ for the selected features.

As a result, a total of $81$ initial datasets, $\mathcal{X}_{init}$ from the original 9 datasets are produced by varying missingness pattern and percentage.

**Data imputation.** For each of the initial datasets $\mathcal{X}_{init}$ with induced missingness, we perform data imputation according to 12 methods ($imp$) and produce a preprocessed dataset for each, $\mathcal{X}_{imp}$. Assume the data is stored such that a row represents an entire datapoint and each column represents a data feature or attribute. The imputation methods are: row removal (i.e., discard all rows with any missing data values), column removal (i.e., remove attribute with missing data values), mean (i.e., replace a missing value with the mean of that attribute), mode (i.e., replace a missing value with the most frequent values within the attribute), $k$-nearest neighbor (KNN) Murti et al. (2019), optimal transport (OT) Muzellec et al. (2020), random sampling (i.e., randomly select samples from the attribute to fill the missing value), multivariate imputation by chained equations (MICE) van Buuren & Groothuis-Oudshoorn (2011), linear interpolation Huang (2021), linear round robin (LRR) Muzellec et al. (2020), MLP round robin Muzellec et al. (2020), and random forest (RF) Hong & Lynn (2020).

### C.3 Data valuation

For each dataset, we encode categorical features into numerical features, and create fixed $80\%/20\%$ train/validation splits. Data splits are maintained across experimental conditions. For TMC-Shapley, G-Shapley, Banzhaf, CS-Shapley and LOO, we use logistic regression model as the learning algorithm $\mathcal{A}$, with $\mathcal{D}$ equal to each dataset's train set; the same applies to fairness computations such as equalized odds difference (EOD). FairShap is computed with $k$NN as the learning algorithm $\mathcal{A}$. The hyperparameters *solver* and *max_iter* were varied for the logistic regression model and value $k$ was varied for the $k$NN neighbor classification model.

Computing TMC-Shapley, G-Shapley, and CS-Shapley data values each required $[4 - 12]$ hours, and Banzhaf and FairShap data values each required $\leq 4$ hours for each dataset $\mathcal{X}_{imp}$. Dataset 18, required 24 hours, due to the larger number of classes (10 classes). TMC-Shapley, G-Shapley, and LOO data values were computed for all datasets. Banzhaf was computed for datasets 23, and 37. CS-Shapley was computed for datasets $18, 23, 31$, and $1680$, each under missingness condition MNAR:30 and on dataset 40994 for all kinds of missingness. FairShap data values were computed for datasets 31 and 1480 for all kinds of missingness. Experiments were conducted using a CPU on a laptop computer with the following hardware specifications: 2.6 GHz 6-Core Intel Core i7 processor; 16 GB 2400 MHz DDR4 RAM; and Intel UHD Graphics 630 1536 MB graphics card.

## D  Metrics for Large-scale Data values Analysis

In this section we develop concise notation (called "conditions") to efficiently report results across a wide range of initial datasets, induced missingness patterns and percentages, imputation methods, and data valuation methods. Conditions are derived as approximate measures of success for data cleaning, class balance, fairness, and group/attribute representation balance, below.

### D.1 Data cleaning definitions

$\texttt{Condition-1A}_j^{tech}$ measures the fraction of datasets for which the data cleaning protocol increases the data value *average*.

$$\texttt{Condition-1A}_j^{tech} = \frac{\sum_{i=1}^{9} \mathbb{1}[avg_{ij}^{tech} > avg_{ir}^{tech}]}{9} \tag{8}$$

$\texttt{Condition-2A}_j^{tech}$ measures the fraction of datasets for which the data cleaning protocol increases the *maximum* data value.

$$\texttt{Condition-1B}_j^{tech} = \frac{\sum_{i=1}^{9} \mathbb{1}[max_{ij}^{tech} > max_{ir}^{tech}]}{9} \tag{9}$$

Here, the term $tech$ denotes the data valuation scheme, $avg$ denotes the average data value, $max$ denotes the maximum data value, and $r$ refers to the baseline dataset condition: the same initial

dataset under row removal imputation. Specifically, the value of $\texttt{Condition-1A}_j^{tech}$ denotes the fraction of datasets for which the average data value after imputing data with algorithm $j$ and valuating with method $tech$ is greater than the average data value after discarding missing data (rows) and valuating with method $tech$. Similarly, the value of $\texttt{Condition-2A}_j^{tech}$ denotes the fraction of datasets for which the maximum data value after imputing data with algorithm $j$ and valuating with method $tech$ is greater than the maximum data value after discarding missing data (rows) and valuating with method $tech$.

## D.2   CLASS BALANCE DEFINITIONS

The class balance $b$ is defined as:

$$b = \begin{cases} \frac{\#minority\ class}{\#majority\ class}, & \text{if } \textit{train-set classes} \geq \textit{test-set classes} \\ 0, & \text{otherwise} \end{cases} \tag{10}$$

$\texttt{Condition-2A}_j^{tech}$ measures the fraction of datasets for which the data subsampling protocol results in a class balance value less than 0.25.

$$\texttt{Condition-2A}_j^{tech} = \frac{\sum_{i=1}^{9} \mathbb{1}[b_{ij}^{tech} < 0.25]}{9} \tag{11}$$

$\texttt{Condition-2B}_j^{tech}$ measures the fraction of datasets for which the data subsampling protocol results in a class balance value less than the original class balance of the unsampled dataset.

$$\texttt{Condition-2B}_j^{tech} = \frac{\sum_{i=1}^{9} \mathbb{1}[b_{ij}^{tech} < b_{ij}]}{9} \tag{12}$$

Here, the term $tech$ denotes the data valuation scheme and $j$ denotes the imputation algorithm. The term $b$ denotes the class balance of the dataset, as computed above. Class balance ($b$) is in the range $[0, 1]$ with zero indicating that at least one class is completely unrepresented in train set, and one indicating that the classes are fully balanced. Specifically, the value of $\texttt{Condition-2A}_j^{tech}$ denotes the fraction of datasets for which the class balance of the dataset subsampled by ranked data value according to data valuation method $tech$ is lower than 0.25. The value of $\texttt{Condition-2B}_j^{tech}$ denotes the fraction of datasets for which the class balance of the dataset subsampled by ranked data value according to data valuation method $tech$ is lower than the class balance of the full "unsampled" dataset.

## D.3   FAIRNESS EQUAL OPPORTUNITY DIFFERENCE (EOD) DEFINITIONS

The fairness measure "equal opportunity difference" (EOD) is defined as:

$$EOD = max(TPR_{diff}, FPR_{diff}) \tag{13}$$

where $TPR_{diff} = |P(\hat{\mathcal{Y}} = 1|\mathcal{Y} = 1, G = 1) - P(\hat{\mathcal{Y}} = 1|\mathcal{Y} = 1, G = 0)|$, and $FPR_{diff} = |P(\hat{\mathcal{Y}} = 1|\mathcal{Y} = 0, G = 1) - P(\hat{\mathcal{Y}} = 1|\mathcal{Y} = 0, G = 0)|$, and $G$ is the sensitive group, and $\hat{\mathcal{Y}}$ is the classifier prediction.

$\texttt{Condition-3}_j^{tech}$ measures whether or not the data subsampling protocol results in an EOD value less than the original EOD of the unsampled dataset; i.e., is 1 if it is "more fair".

$$\texttt{Condition-3}_j^{tech} = \mathbb{1}[EOD_j^{tech} < EOD_j] \tag{14}$$

Here, the term $tech$ denotes the data valuation technique, $j$ denotes the imputation algorithm and $EOD$ denotes the equalized odds difference ($EOD$) as defined above. When the value of $\texttt{Condition-3}_j^{tech}$ is 1, it implies that the $EOD$ of the dataset subsampled by ranked data value according to data valuation method $tech$ is lower than the $EOD$ of the full "unsampled" dataset. Value 0 implies the reverse. Since lower $EOD$ implies better model fairness, a value of 1 is more desirable in this scenario.

### D.4 GROUP AND ATTRIBUTE REPRESENTATION BALANCE DEFINITIONS

The group (or attribute) representation balance $g$ is defined as:

$$g = \frac{\#minority\ subgroup}{\#majority\ subgroup} \tag{15}$$

$\texttt{Condition-4}_j^{tech}$ measures whether or not the data subsampling protocol results in a group (or attribute) representation balance value less than the original balance value of the unsampled dataset; i.e., is 1 if it is "less balanced".

$$\texttt{Condition-4}_j^{tech} = \mathbb{1}[g_j^{tech} < g_j] \tag{16}$$

Here, the term $tech$ denotes the data valuation technique, $j$ denotes the imputation algorithm used and $g$ denotes the group representation balance described above. For example, if the group is "binary sex", then the subgroups could be "male" and "female". When the value of $\texttt{Condition-4}_j^{tech}$ is 1, it implies that the group (or attribute) representation balance of the dataset subsampled by ranked data value according to data valuation method $tech$ is lower than the balance of the full "unsampled" dataset. Value 0 implies the reverse. A value of 1 is more desirable in this scenario.

# E  DATA PREPROCESSING CAN DRASTICALLY ALTER DATA VALUES

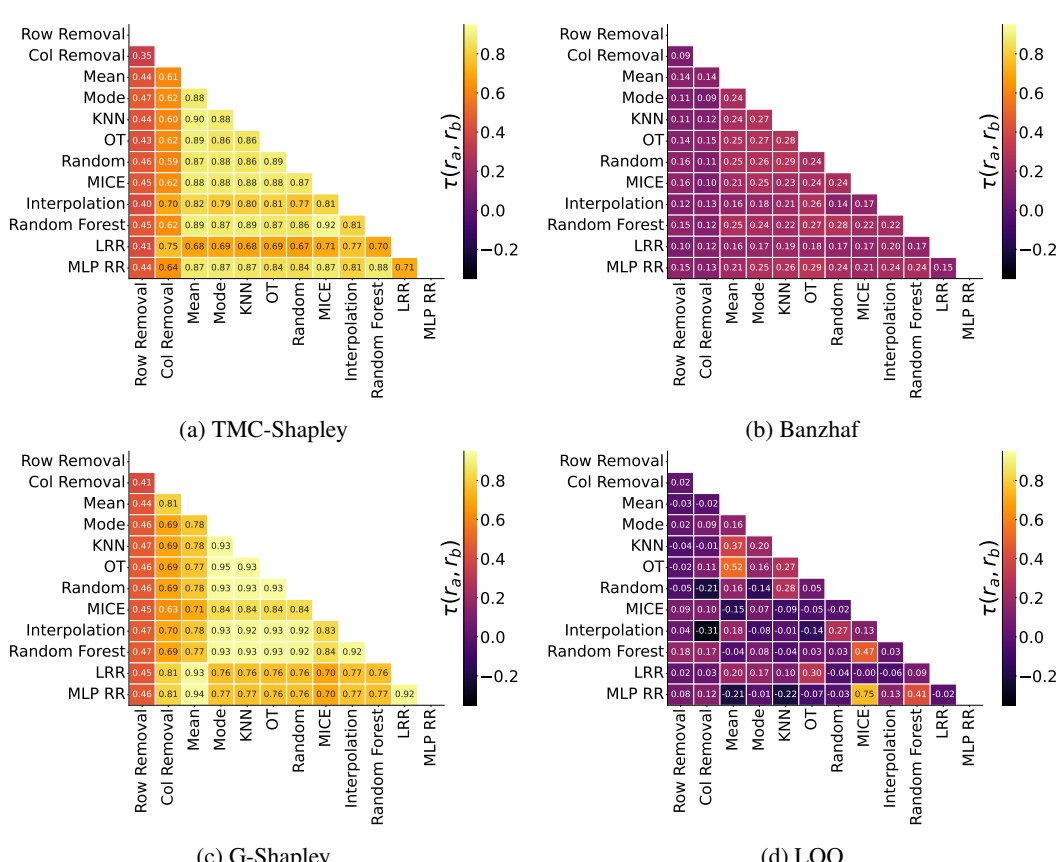

Figure 5: The Kendall tau values when cross-comparing the data ranks resulting from imputation algorithms on dataset 37 (Pima Indians Diabetes Database, with MNAR-10). The observed tau values for (**a**) TMC-Shapley, (**b**) Banzhaf, and (**c**) G-Shapley are typically $> 0$ and $< 1$ indicating a positive correlation between the compared ranks. However, for (**d**) LOO the tau values are usually $< 0$ indicating a negative correlation and high disagreement between rank orders.

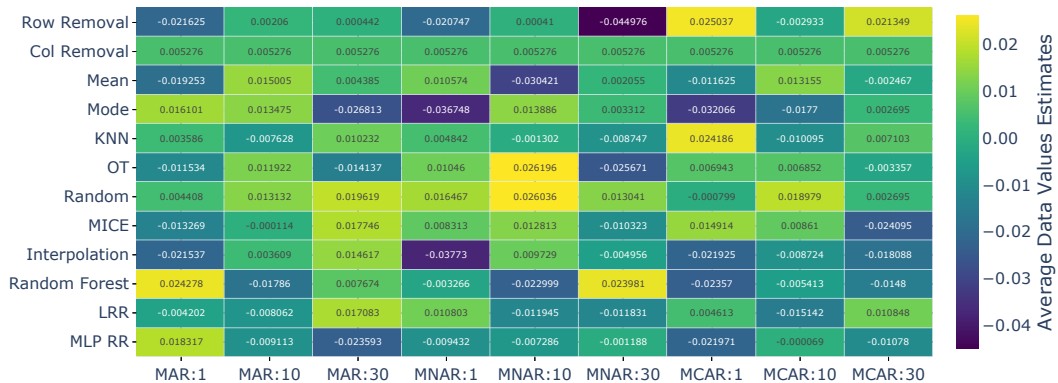

Figure 6: Average LOO data values for dataset 1063 (KC2 Software defect prediction), varied by missingness pattern/percentage and imputation method.

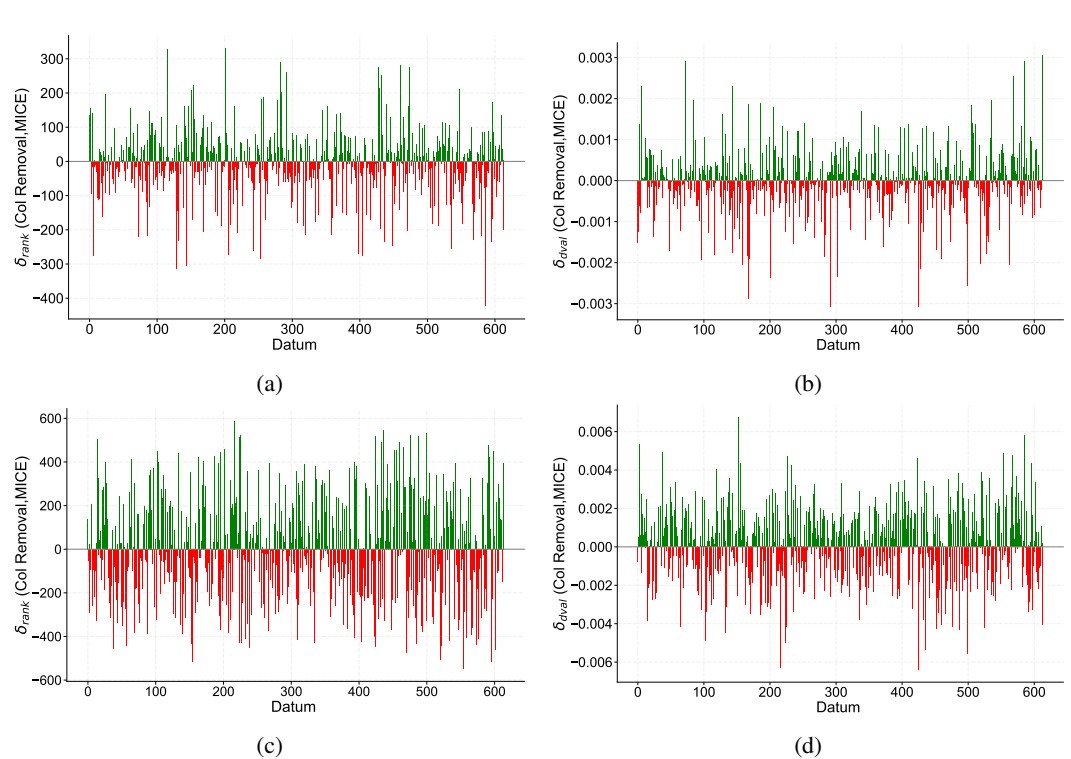

Figure 7: Changes in (**a**) TMC-Shapley rank order and (**b**) data values, and (**c**) Banzhaf rank order and (**d**) data values for individual data points across applications of column removal and MICE imputation methods on dataset 37 (Pima Indians Diabetes Database, with MNAR-10).

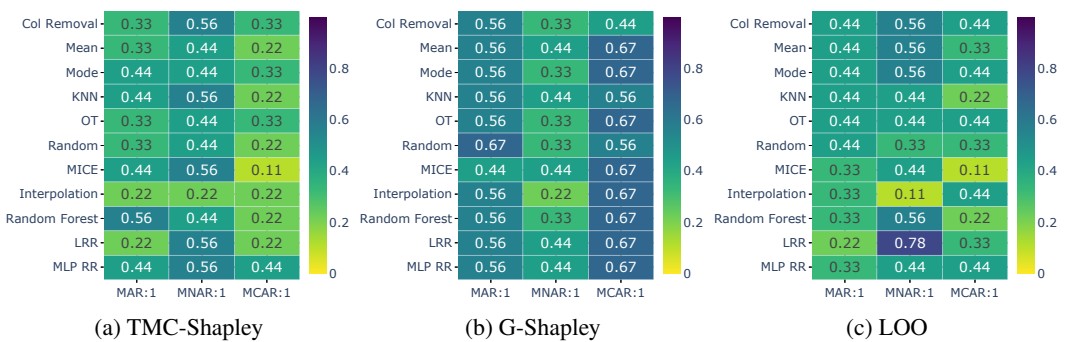

Figure 8: $\texttt{Condition-1A}_j^{tech}$ on three data valuation methods, which measures the fraction of datasets for which the data cleaning protocol increases the data value *average* compared to a baseline method (see Appendix Section D). Fractions are shown for (**a**) $\texttt{Condition-1A}_j^{TMC-Shapley}$, (**b**) $\texttt{Condition-1A}_j^{G-Shapley}$ and (**c**) $\texttt{Condition-1A}_j^{LOO}$ across all datasets and missingness MAR:1, MNAR:1 and MNAR:1 applied. For TMC-Shapley and LOO, most imputation algorithms resulted in a lower average data value than the baseline method; the opposite was true for G-Shapley.

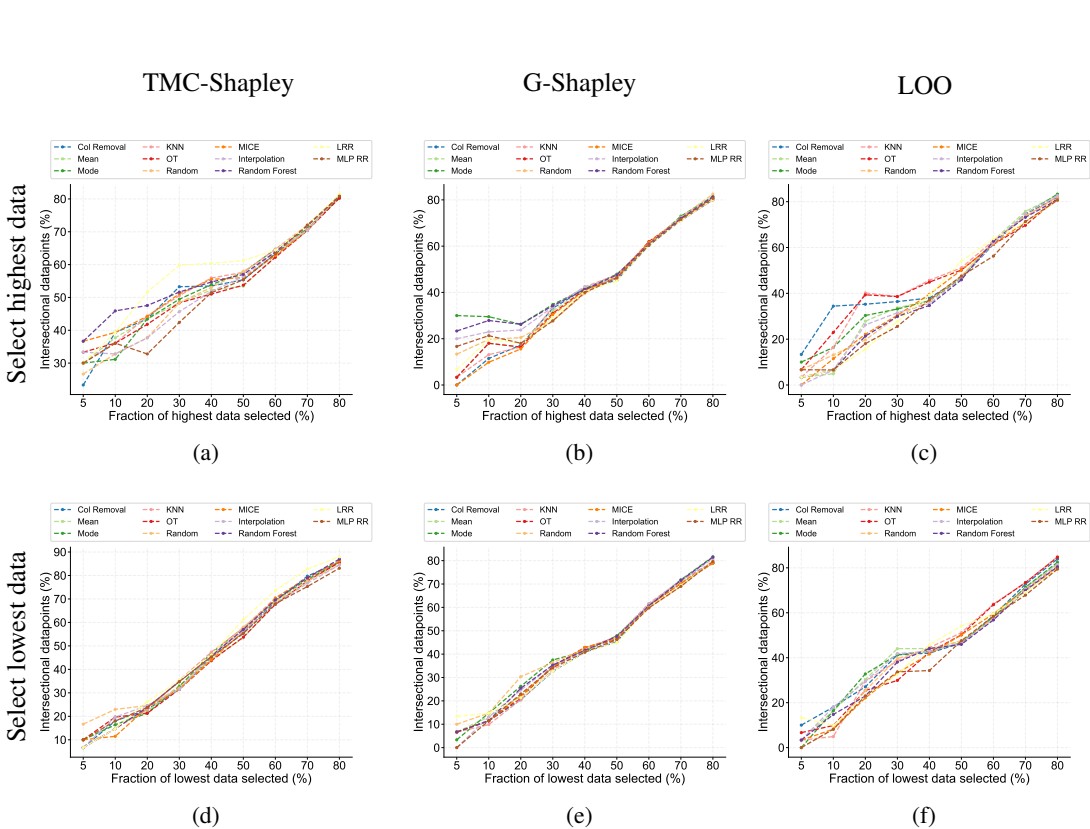

Figure 9: The percentage of shared points between high- and low- data value sets as a function of various imputation methods and the baseline method (row removal). Across all data valuation methods ((**a,d**) TMC-Shapley, (**b,e**) G-Shapley, and (**c,f**) LOO), higher variance is observed in intersectional points when excluding low-valued data (**a-c**) than excluding high-valued data (**d-f**).

|  | MAR:1 | MAR:10 | MAR:30 | MNAR:1 | MNAR:10 | MNAR:30 | MCAR:1 | MCAR:10 | MCAR:30 |
|---|---|---|---|---|---|---|---|---|---|
| Col Removal | (1/3,5/9,4/9) | (1/9,4/9,2/3) | (1/3,2/9,7/9) | (5/9,1/3,5/9) | (1/3,5/9,2/3) | (2/9,1/3,4/9) | (1/3,4/9,4/9) | (1/3,1/3,2/3) | (2/9,2/9,2/3) |
| Mean | (1/3,5/9,4/9) | (1/9,2/3,2/3) | (1/3,4/9,2/3) | (4/9,4/9,5/9) | (1/3,4/9,2/3) | (2/9,1/3,5/9) | (2/9,2/3,1/3) | (1/3,5/9,4/9) | (2/9,1/3,1/3) |
| Mode | (4/9,5/9,4/9) | (1/9,5/9,5/9) | (1/3,1/3,2/3) | (4/9,1/3,5/9) | (1/3,4/9,4/9) | (2/9,1/3,1/3) | (1/3,2/3,4/9) | (2/9,5/9,4/9) | (2/9,4/9,4/9) |
| KNN | (4/9,5/9,4/9) | (1/9,5/9,4/9) | (1/3,1/3,2/3) | (5/9,4/9,4/9) | (1/3,4/9,2/3) | (2/9,1/3,5/9) | (2/9,5/9,2/9) | (2/9,5/9,1/3) | (2/9,2/9,4/9) |
| OT | (1/3,5/9,4/9) | (1/9,2/3,2/3) | (1/3,4/9,4/9) | (4/9,1/3,4/9) | (1/3,4/9,5/9) | (2/9,1/3,4/9) | (1/3,2/3,4/9) | (1/3,5/9,1/3) | (2/9,1/3,4/9) |
| Random | (1/3,2/3,4/9) | (1/9,5/9,5/9) | (1/3,4/9,5/9) | (4/9,1/3,1/3) | (1/3,4/9,5/9) | (2/9,1/3,5/9) | (2/9,5/9,1/3) | (1/3,5/9,4/9) | (2/9,1/3,2/9) |
| MICE | (4/9,4/9,1/3) | (1/9,5/9,4/9) | (1/3,4/9,7/9) | (5/9,4/9,4/9) | (1/3,4/9,2/3) | (2/9,1/3,4/9) | (1/9,2/3,1/9) | (1/3,5/9,4/9) | (1/3,4/9,4/9) |
| Interpolation | (2/9,5/9,1/3) | (1/9,5/9,2/3) | (1/3,1/3,5/9) | (2/9,2/9,1/9) | (1/3,4/9,2/3) | (2/9,1/3,5/9) | (2/9,2/3,4/9) | (1/3,2/3,2/9) | (1/3,4/9,1/3) |
| Random Forest | (5/9,5/9,1/3) | (1/9,5/9,5/9) | (1/3,1/3,5/9) | (4/9,1/3,5/9) | (1/3,4/9,5/9) | (2/9,1/3,5/9) | (2/9,2/3,2/9) | (2/9,4/9,5/9) | (1/3,2/9,5/9) |
| LRR | (2/9,5/9,2/9) | (1/9,2/3,1/3) | (1/3,1/9,2/3) | (5/9,4/9,7/9) | (1/3,4/9,1/3) | (2/9,1/3,5/9) | (2/9,2/3,1/3) | (1/9,5/9,1/3) | (2/9,2/9,1/3) |
| MLP RR | (4/9,5/9,1/3) | (1/9,2/3,4/9) | (2/9,2/9,5/9) | (5/9,4/9,4/9) | (1/3,4/9,5/9) | (1/3,1/3,4/9) | (4/9,2/3,4/9) | (4/9,5/9,1/3) | (2/9,1/3,5/9) |

(a) $\text{Condition-1A}_j^{tech}$ across 3 data valuation methods, 11 imputation methods, and 9 missingness conditions; this measures the fraction of datasets for which the data cleaning protocol increases the data value *average* compared to a baseline method (see Appendix Section D). Each cell value denotes a triplet of results for the three data valuation techniques: ($\text{Condition-1A}_j^{TMC-Shapley}$, $\text{Condition-1A}_j^{G-Shapley}$, $\text{Condition-1A}_j^{LOO}$), where $j$ is the imputation algorithm. The highlighted values in blue denote settings where handling missing data improves **average** data value for majority of the datasets.

|  | MAR:1 | MAR:10 | MAR:30 | MNAR:1 | MNAR:10 | MNAR:30 | MCAR:1 | MCAR:10 | MCAR:30 |
|---|---|---|---|---|---|---|---|---|---|
| Col Removal | (4/9,2/9,2/9) | (0/9,5/9,1/3) | (0/9,2/9,4/9) | (5/9,5/9,5/9) | (0/9,1/3,4/9) | (0/9,1/9,5/9) | (1/3,5/9,1/3) | (1/3,1/3,5/9) | (1/9,2/9,1/3) |
| Mean | (5/9,2/9,1/3) | (1/9,1/3,4/9) | (0/9,2/9,2/9) | (4/9,4/9,4/9) | (0/9,2/9,4/9) | (1/9,1/9,1/3) | (2/9,2/9,1/3) | (4/9,2/9,2/3) | (2/9,1/3,2/9) |
| Mode | (2/3,2/9,4/9) | (1/9,1/3,1/3) | (0/9,1/3,1/9) | (2/3,4/9,1/9) | (1/9,2/9,4/9) | (1/9,1/9,1/3) | (2/9,1/3,1/3) | (2/9,1/3,2/9) | (1/3,2/9,1/9) |
| KNN | (5/9,2/9,2/9) | (0/9,1/3,1/3) | (0/9,2/9,2/9) | (2/3,4/9,2/9) | (1/9,2/9,5/9) | (0/9,0/9,4/9) | (1/9,2/9,1/9) | (1/3,2/9,2/9) | (1/9,1/3,2/9) |
| OT | (5/9,1/9,2/9) | (1/9,1/3,1/3) | (0/9,2/9,1/3) | (5/9,4/9,2/9) | (0/9,2/9,4/9) | (1/9,1/9,4/9) | (1/3,2/9,1/3) | (4/9,2/9,4/9) | (2/9,2/9,1/9) |
| Random | (5/9,1/9,4/9) | (0/9,4/9,2/9) | (0/9,2/9,1/3) | (2/3,4/9,2/9) | (2/9,1/3,2/3) | (1/9,0/9,5/9) | (2/9,1/3,2/9) | (4/9,2/9,5/9) | (2/9,1/3,2/9) |
| MICE | (2/3,2/9,1/9) | (0/9,1/3,2/9) | (0/9,2/9,4/9) | (2/3,4/9,2/9) | (0/9,2/9,1/3) | (1/9,1/9,4/9) | (2/9,1/3,0/9) | (1/3,2/9,1/3) | (1/9,1/3,1/3) |
| Interpolation | (5/9,2/9,1/9) | (1/9,4/9,1/3) | (0/9,1/3,2/9) | (1/9,2/9,2/9) | (1/9,2/9,5/9) | (1/9,0/9,4/9) | (2/9,1/3,1/3) | (5/9,1/9,1/3) | (2/9,2/9,2/9) |
| Random Forest | (5/9,2/9,2/9) | (0/9,1/3,1/3) | (0/9,1/3,2/9) | (4/9,4/9,2/9) | (0/9,2/9,4/9) | (1/9,2/9,4/9) | (2/9,2/9,1/9) | (2/9,1/3,4/9) | (2/9,1/3,1/9) |
| LRR | (2/3,2/9,1/3) | (1/9,1/3,4/9) | (2/9,1/3,5/9) | (2/3,4/9,4/9) | (0/9,2/9,4/9) | (1/9,1/9,5/9) | (1/3,1/3,2/9) | (1/9,2/9,2/9) | (2/9,1/3,2/9) |
| MLP RR | (8/9,2/9,5/9) | (1/9,4/9,4/9) | (2/9,1/3,2/9) | (5/9,4/9,1/3) | (1/9,2/9,1/3) | (0/9,1/9,4/9) | (1/9,1/9,1/9) | (2/9,2/9,1/3) | (2/9,1/3,4/9) |

(b) $\text{Condition-1B}_j^{tech}$ across 3 data valuation methods, 11 imputation methods, and 9 missingness conditions; this measures the fraction of datasets for which the data cleaning protocol increases the data value *maximum* compared to a baseline method (see Appendix Section D). Each cell value denotes a triplet of results for the three data valuation techniques: ($\text{Condition-1B}_j^{TMC-Shapley}$, $\text{Condition-1B}_j^{G-Shapley}$, $\text{Condition-1B}_j^{LOO}$), where $j$ is the imputation algorithm. The highlighted values in blue denote cases in which handling missing data improves the **maximum** data value for majority of the datasets.

Table 2: $\text{Condition-1A}_j^{tech}$ and $\text{Condition-1B}_j^{tech}$. Handling missing values generally improves the (**a**) **average** data value, and in some cases, the (**b**) **maximum** data value.

# F   DATA VALUE BASED SUBSAMPLING CAN INCREASE CLASS IMBALANCE

Figure 10: The distribution of TMC-Shapley, G-Shapley and LOO data values according to target class. Distributions are shown for: (*a-c*) dataset 23 (Contraceptive method choice, MNAR-30), (*d-f*) 18 (Mfeat-morphological, MNAR-30) and (*g-i*) 40994 (climate-model-simulation-crashes, MCAR-10). Under certain conditions, e.g. (**b**) and (**h**), strong class bias exists in data values, as evidenced by disparate distributions by class. In these cases, data sampling according to data value would likely result in greater amounts of data excluded from specific classes.

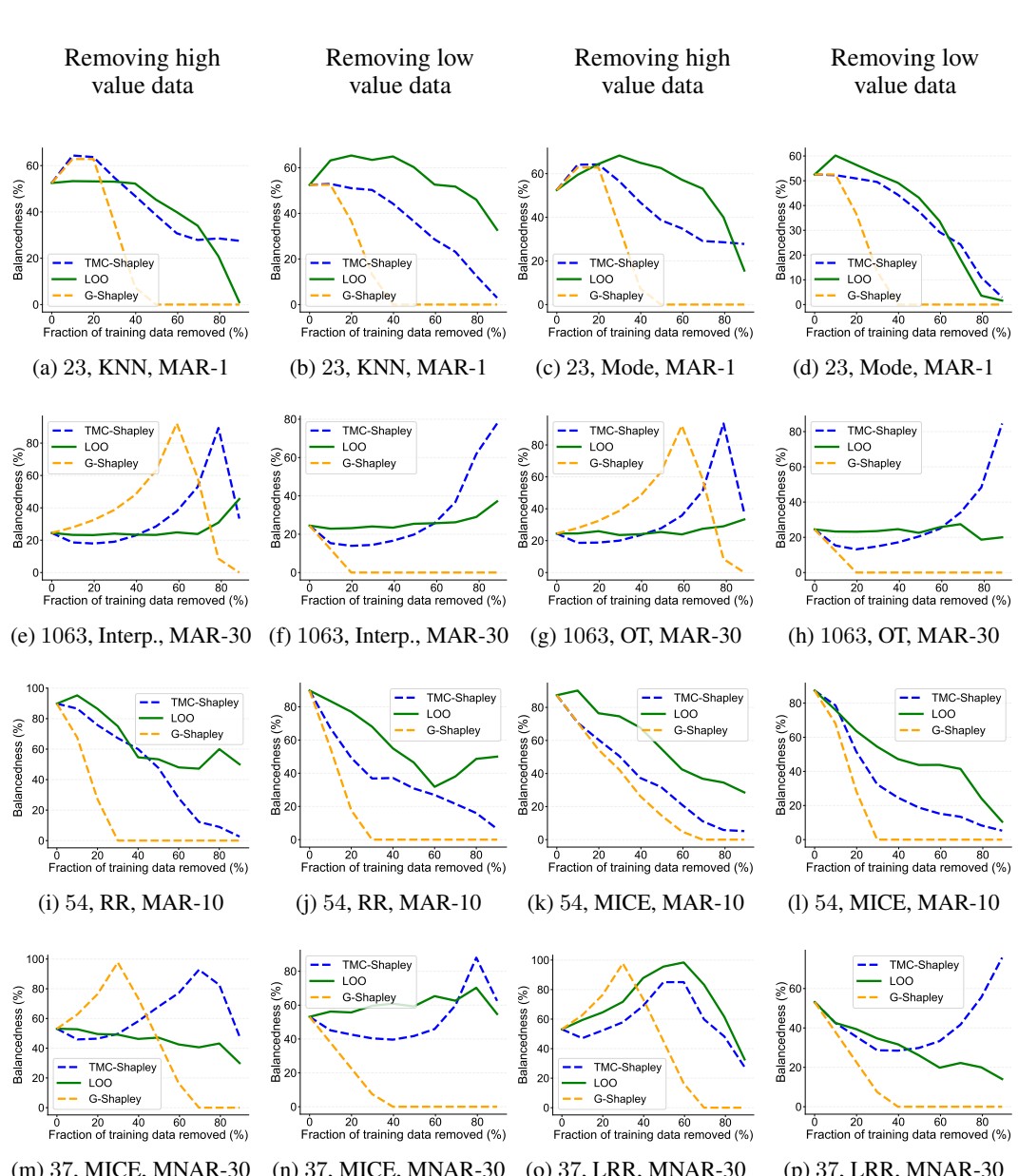

Figure 11: Class balance (as defined in Appendix D as *b*) versus percentage of data removed, as a function of four data valuation metrics (TMC-Shapley, G-Shapley, LOO and CS-Shapley). Subfigure captions indicate the dataset, imputation method, and missingness pattern/percentage. These factors have varied effects on the class balance.

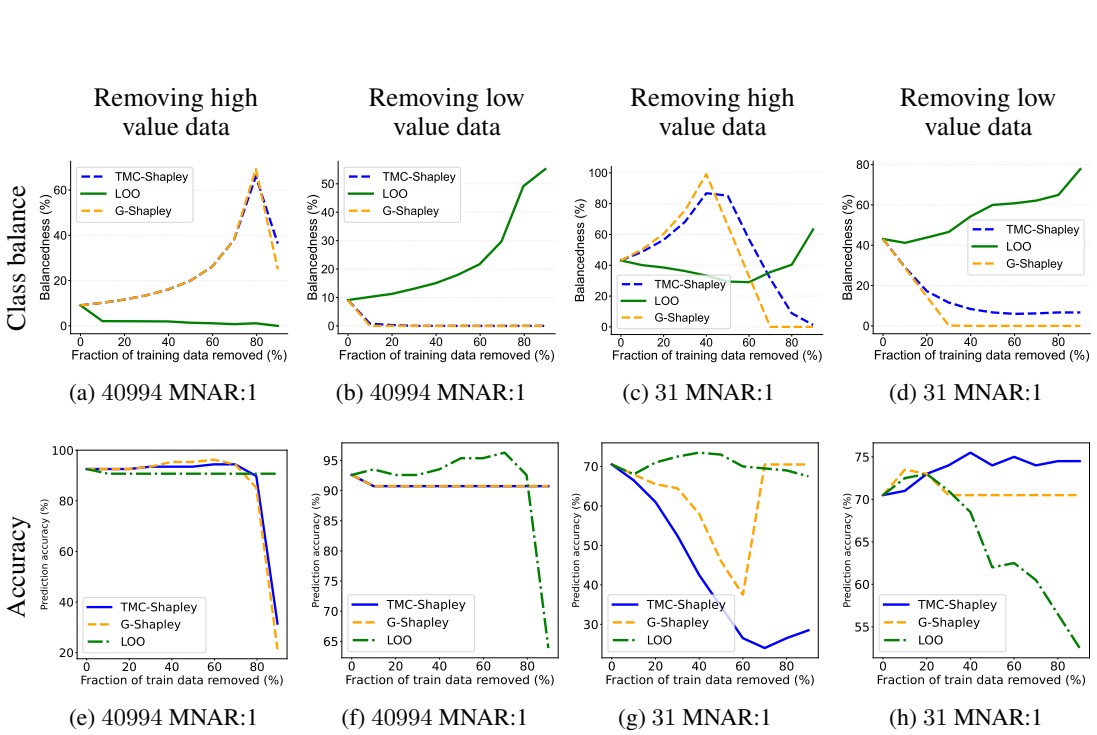

Figure 12: (**a-d**) Class balance and (**e-h**) model prediction accuracy as a function of subsampling for four experimental conditions. Subfigure captions list dataset and missingness pattern/percentage; imputation method is column removal for each case. We observe a relationship between the unsampled dataset class imlabance and the effects on class balance following value-based subsampling and accuracy. In datasets with low initial class balance, e.g. dataset 40994 (climate-model-simulation-crashes, (**a,b,e,f**)), the removal of high-value data via TMC- and G-Shapley initially (**a**) increases class balance, while removal of low-value data via TMC- and G-Shapley initially (**b**) decreases class balance. Correspondingly, the accuracy of the model trained on these subsampled datasets initially (**e**) increases and (**f**) decreases, respectively. Datasets with higher initial class balance, e.g. dataset 31 (German credit, (**c,d,g,h**)) tend to exhibit more drastic changes to prediction accuracy as a function of subsampling.

| | MAR:1 | MAR:10 | MAR:30 | MNAR:1 | MNAR:10 | MNAR:30 | MCAR:1 | MCAR:10 | MCAR:30 |
|---|---|---|---|---|---|---|---|---|---|
| Row Removal | (2/3,7/9,1/3) | (2/3,7/9,1/3) | (5/9,7/9,1/3) | (2/3,7/9,1/3) | (5/9,2/3,4/9) | (2/3,7/9,5/9) | (5/9,7/9,1/3) | (2/3,7/9,1/3) | (2/3,7/9,1/3) |
| Col Removal | (2/3,7/9,1/3) | (2/3,7/9,1/3) | (2/3,7/9,1/3) | (2/3,7/9,1/3) | (2/3,7/9,1/3) | (2/3,7/9,1/3) | (2/3,7/9,1/3) | (2/3,7/9,1/3) | (2/3,7/9,1/3) |
| Mean | (2/3,7/9,1/3) | (5/9,7/9,1/3) | (2/3,7/9,1/3) | (5/9,7/9,2/9) | (5/9,7/9,1/3) | (2/3,7/9,1/3) | (5/9,7/9,1/3) | (2/3,7/9,1/3) | (2/3,7/9,1/3) |
| Mode | (5/9,7/9,1/3) | (5/9,7/9,1/3) | (5/9,7/9,1/3) | (2/3,7/9,1/3) | (5/9,7/9,1/3) | (2/3,7/9,4/9) | (5/9,7/9,1/3) | (5/9,7/9,1/3) | (5/9,7/9,1/3) |
| KNN | (5/9,7/9,1/3) | (5/9,7/9,1/3) | (5/9,7/9,1/3) | (2/3,7/9,1/3) | (2/3,7/9,4/9) | (2/3,7/9,2/9) | (2/3,7/9,1/3) | (5/9,7/9,1/3) | (2/3,7/9,1/3) |
| OT | (5/9,7/9,1/3) | (5/9,7/9,4/9) | (2/3,7/9,1/3) | (5/9,7/9,1/3) | (5/9,7/9,1/3) | (5/9,7/9,1/3) | (5/9,7/9,1/3) | (2/3,7/9,1/3) | (5/9,7/9,4/9) |
| random | (5/9,7/9,1/3) | (5/9,7/9,1/3) | (2/3,7/9,1/3) | (5/9,7/9,1/3) | (2/3,7/9,1/3) | (5/9,7/9,1/3) | (5/9,7/9,1/3) | (2/3,7/9,4/9) | (2/3,7/9,4/9) |
| MICE | (5/9,7/9,1/3) | (5/9,7/9,1/3) | (5/9,7/9,1/3) | (5/9,7/9,1/3) | (5/9,7/9,1/3) | (5/9,7/9,1/3) | (5/9,7/9,1/3) | (5/9,7/9,1/3) | (5/9,7/9,1/3) |
| Interpolation | (5/9,7/9,1/3) | (5/9,7/9,4/9) | (2/3,7/9,1/3) | (5/9,7/9,1/3) | (2/3,7/9,1/3) | (5/9,7/9,2/9) | (5/9,7/9,4/9) | (2/3,7/9,1/3) | (2/3,7/9,4/9) |
| Random Forest | (5/9,7/9,1/3) | (5/9,7/9,1/3) | (5/9,7/9,1/3) | (5/9,7/9,2/9) | (5/9,7/9,4/9) | (5/9,7/9,1/3) | (5/9,7/9,1/3) | (5/9,7/9,1/3) | (2/3,7/9,1/3) |
| LRR | (5/9,7/9,1/3) | (2/3,7/9,4/9) | (2/3,7/9,4/9) | (5/9,7/9,1/3) | (5/9,7/9,1/3) | (5/9,7/9,1/3) | (2/3,7/9,1/3) | (2/3,7/9,2/9) | (2/3,7/9,4/9) |
| MLP RR | (5/9,7/9,1/3) | (2/3,7/9,1/3) | (2/3,7/9,4/9) | (5/9,7/9,2/9) | (5/9,7/9,1/3) | (2/3,7/9,4/9) | (5/9,7/9,1/3) | (2/3,7/9,1/3) | (2/3,7/9,4/9) |

(a) $\texttt{Condition-2A}_j^{tech}$ across 3 data valuation methods, 12 imputation methods, and 9 missingness conditions; this measures the fraction of datasets for which the data subsampling protocol results in a class balance value less than 0.25 (see Appendix Section D). Each cell value denotes a triplet of results for the three data valuation techniques: ($\texttt{Condition-2A}_j^{TMC-Shapley}$, $\texttt{Condition-2A}_j^{G-Shapley}$, $\texttt{Condition-2A}_j^{LOO}$), where $j$ denotes the imputation algorithm used on the data. Results are specific to when the subsampled data is 80% of **highest value data**. The highlighted values in teal color denote experimental conditions for which subsampled data has a class balance greater than 0.25 for the majority of the datasets. Subsampling with TMC-Shapley and G-Shapley generally results in class balance worse than 0.25.

| | MAR:1 | MAR:10 | MAR:30 | MNAR:1 | MNAR:10 | MNAR:30 | MCAR:1 | MCAR:10 | MCAR:30 |
|---|---|---|---|---|---|---|---|---|---|
| Row Removal | (9/9,9/9,5/9) | (8/9,9/9,4/9) | (7/9,9/9,2/9) | (9/9,9/9,5/9) | (8/9,9/9,7/9) | (7/9,9/9,8/9) | (9/9,9/9,1/3) | (8/9,9/9,4/9) | (9/9,9/9,2/3) |
| Col Removal | (9/9,9/9,4/9) | (9/9,9/9,4/9) | (9/9,9/9,4/9) | (9/9,9/9,4/9) | (9/9,9/9,4/9) | (9/9,9/9,4/9) | (9/9,9/9,4/9) | (9/9,9/9,4/9) | (9/9,9/9,4/9) |
| Mean | (9/9,9/9,2/3) | (9/9,9/9,5/9) | (9/9,9/9,2/3) | (9/9,9/9,1/3) | (9/9,9/9,2/3) | (9/9,9/9,4/9) | (9/9,9/9,5/9) | (9/9,9/9,5/9) | (9/9,9/9,7/9) |
| Mode | (9/9,9/9,5/9) | (9/9,9/9,5/9) | (9/9,9/9,5/9) | (9/9,9/9,2/3) | (9/9,9/9,7/9) | (9/9,9/9,5/9) | (9/9,9/9,5/9) | (9/9,9/9,4/9) | (9/9,9/9,8/9) |
| KNN | (9/9,9/9,5/9) | (9/9,9/9,7/9) | (9/9,9/9,2/3) | (9/9,9/9,5/9) | (9/9,9/9,1/3) | (9/9,9/9,2/3) | (9/9,9/9,4/9) | (9/9,9/9,7/9) | (9/9,9/9,5/9) |
| OT | (9/9,9/9,5/9) | (9/9,9/9,1/3) | (9/9,9/9,2/3) | (9/9,9/9,4/9) | (9/9,9/9,5/9) | (9/9,9/9,2/3) | (9/9,9/9,5/9) | (9/9,9/9,2/3) | (9/9,9/9,8/9) |
| random | (9/9,9/9,2/3) | (9/9,9/9,5/9) | (9/9,9/9,4/9) | (9/9,9/9,5/9) | (9/9,9/9,5/9) | (9/9,9/9,5/9) | (9/9,9/9,5/9) | (9/9,9/9,4/9) | (9/9,9/9,2/3) |
| MICE | (9/9,9/9,1/3) | (9/9,9/9,2/3) | (9/9,9/9,2/3) | (9/9,9/9,5/9) | (9/9,9/9,5/9) | (9/9,9/9,5/9) | (9/9,9/9,5/9) | (9/9,9/9,5/9) | (9/9,9/9,7/9) |
| Interpolation | (9/9,9/9,2/3) | (9/9,9/9,5/9) | (9/9,9/9,4/9) | (9/9,9/9,5/9) | (9/9,9/9,5/9) | (9/9,9/9,1/3) | (9/9,9/9,5/9) | (9/9,9/9,2/3) | (9/9,9/9,2/3) |
| Random Forest | (9/9,9/9,4/9) | (9/9,9/9,7/9) | (9/9,9/9,4/9) | (9/9,9/9,4/9) | (9/9,9/9,4/9) | (9/9,9/9,2/3) | (9/9,9/9,1/3) | (9/9,9/9,5/9) | (9/9,9/9,2/3) |
| LRR | (9/9,9/9,1/3) | (9/9,9/9,2/3) | (8/9,9/9,4/9) | (9/9,9/9,1/3) | (9/9,9/9,5/9) | (9/9,9/9,2/3) | (9/9,9/9,2/3) | (9/9,9/9,1/3) | (9/9,9/9,2/3) |
| MLP RR | (9/9,9/9,5/9) | (9/9,9/9,7/9) | (9/9,9/9,5/9) | (9/9,9/9,4/9) | (9/9,9/9,5/9) | (9/9,9/9,5/9) | (9/9,9/9,5/9) | (9/9,9/9,7/9) | (9/9,9/9,2/3) |

(b) $\texttt{Condition-2B}_j^{tech}$ across 3 data valuation methods, 12 imputation methods, and 9 missingness conditions; this measures the fraction of datasets for which the data subsampling protocol results in a class balance value less than the original class balance of the unsampled dataset (see Appendix Section D). Each cell value denotes a triplet of results for the three data valuation techniques: ($\texttt{Condition-2B}_j^{TMC-Shapley}$, $\texttt{Condition-2B}_j^{G-Shapley}$, $\texttt{Condition-2B}_j^{LOO}$), where $j$ denotes the imputation algorithm used on the data. Results are specific to experimental conditions in which the subsampled data is 80% of **highest value data**. The highlighted values in teal color denote settings where subsampled data has a class balance greater than the full "unsampled" data, for the majority of the datasets. Across all conditions, subsampling generally via any data valuation method generally results in worse class balance than the unsampled set.

Table 3: $\texttt{Condition-2A}_j^{tech}$ and $\texttt{Condition-2B}_j^{tech}$ on datasets subsampled by selecting the highest value data (80%). Generally, class balance worsens due to subsampling, both (**a**) in overall class balance scores ($b$ less than 0.25), and (**b**) relatively with respect to the unsampled dataset.

| | MAR:1 | MAR:10 | MAR:30 | MNAR:1 | MNAR:10 | MNAR:30 | MCAR:1 | MCAR:10 | MCAR:30 |
|---|---|---|---|---|---|---|---|---|---|
| Row Removal | (1/3,4/9,1/3) | (4/9,4/9,1/3) | (4/9,1/3,1/3) | (1/3,4/9,1/3) | (4/9,1/3,1/3) | (4/9,5/9,4/9) | (1/3,4/9,2/9) | (4/9,4/9,2/9) | (1/3,4/9,1/3) |
| Col Removal | (4/9,4/9,1/3) | (4/9,4/9,1/3) | (4/9,4/9,1/3) | (4/9,4/9,1/3) | (4/9,4/9,1/3) | (4/9,4/9,1/3) | (4/9,4/9,1/3) | (4/9,4/9,1/3) | (4/9,4/9,1/3) |
| Mean | (1/3,1/3,1/3) | (1/3,1/3,1/3) | (4/9,1/3,2/9) | (1/3,1/3,2/9) | (1/3,1/3,1/3) | (1/3,1/3,1/3) | (1/3,1/3,2/9) | (1/3,1/3,4/9) | (1/3,1/3,1/3) |
| Mode | (1/3,1/3,1/3) | (4/9,1/3,1/3) | (4/9,4/9,4/9) | (1/3,1/3,4/9) | (4/9,1/3,2/9) | (1/3,1/3,1/3) | (1/3,1/3,2/9) | (4/9,1/3,1/3) | (1/3,1/3,4/9) |
| KNN | (1/3,1/3,2/9) | (1/3,1/3,1/3) | (1/3,1/3,1/3) | (1/3,1/3,2/9) | (1/3,1/3,2/9) | (1/3,1/3,1/3) | (1/3,1/3,1/3) | (1/3,1/3,1/3) | (1/3,1/3,1/3) |
| OT | (1/3,1/3,1/3) | (1/3,1/3,1/3) | (4/9,1/3,2/9) | (1/3,1/3,2/9) | (1/3,1/3,2/9) | (1/3,1/3,1/3) | (1/3,1/3,1/3) | (1/3,1/3,4/9) | (1/3,1/3,4/9) |
| random | (1/3,1/3,1/3) | (4/9,1/3,2/9) | (1/3,1/3,1/3) | (1/3,1/3,1/3) | (1/3,4/9,1/3) | (1/3,4/9,1/3) | (1/3,1/3,1/3) | (1/3,1/3,1/3) | (1/3,1/3,4/9) |
| MICE | (1/3,1/3,2/9) | (1/3,1/3,1/3) | (1/3,1/3,1/3) | (1/3,1/3,1/3) | (1/3,1/3,1/3) | (1/3,1/3,2/9) | (1/3,1/3,4/9) | (1/3,1/3,2/9) | (1/3,1/3,2/9) |
| Interpolation | (1/3,1/3,2/9) | (4/9,1/3,1/3) | (4/9,1/3,1/3) | (1/3,1/3,4/9) | (1/3,1/3,2/9) | (1/3,1/3,1/3) | (1/3,1/3,2/9) | (1/3,1/3,1/3) | (1/3,1/3,2/9) |
| Random Forest | (1/3,1/3,1/3) | (1/3,1/3,4/9) | (1/3,1/3,4/9) | (1/3,1/3,5/9) | (1/3,1/3,1/3) | (1/3,1/3,1/3) | (1/3,1/3,4/9) | (1/3,1/3,2/9) | (1/3,1/3,1/3) |
| LRR | (1/3,1/3,2/9) | (4/9,1/3,2/9) | (1/3,1/3,2/9) | (1/3,1/3,1/3) | (4/9,1/3,4/9) | (1/3,1/3,1/3) | (4/9,1/3,1/3) | (1/3,1/3,1/3) | (1/3,1/3,1/3) |
| MLP RR | (1/3,1/3,1/3) | (1/3,1/3,2/9) | (4/9,1/3,2/9) | (1/3,1/3,1/3) | (1/3,1/3,2/9) | (1/3,1/3,1/3) | (1/3,1/3,1/3) | (1/3,1/3,1/3) | (1/3,1/3,2/9) |

(a) `Condition-2A`$_j^{tech}$ across 3 data valuation methods, 12 imputation methods, and 9 missingness conditions; this measures the fraction of datasets for which the data subsampling protocol results in a class balance value less than 0.25 (see Appendix Section D). Each cell value denotes a triplet of results for the three data valuation techniques: (`Condition-2A`$_j^{TMC-Shapley}$, `Condition-2A`$_j^{G-Shapley}$, `Condition-2A`$_j^{LOO}$), where $j$ denotes the imputation algorithm used on the data. Results are specific to when the subsampled data is 80% of **lowest value data**. The highlighted values in teal color denote experimental conditions for which subsampled data has a class balance greater than 0.25 for the majority of the datasets. Excluding high-value data using any data valuation metric generally results in class balance greater than 0.25.

| | MAR:1 | MAR:10 | MAR:30 | MNAR:1 | MNAR:10 | MNAR:30 | MCAR:1 | MCAR:10 | MCAR:30 |
|---|---|---|---|---|---|---|---|---|---|
| Row Removal | (4/9,2/9,2/3) | (1/3,2/9,9/9) | (4/9,2/9,8/9) | (4/9,2/9,5/9) | (4/9,2/9,4/9) | (4/9,2/9,2/3) | (4/9,2/9,5/9) | (4/9,2/9,2/3) | (4/9,2/9,1/3) |
| Col Removal | (4/9,2/9,7/9) | (4/9,2/9,7/9) | (4/9,2/9,7/9) | (4/9,2/9,7/9) | (4/9,2/9,7/9) | (4/9,2/9,7/9) | (4/9,2/9,7/9) | (4/9,2/9,7/9) | (4/9,2/9,7/9) |
| Mean | (4/9,2/9,4/9) | (4/9,2/9,7/9) | (4/9,2/9,4/9) | (4/9,2/9,4/9) | (4/9,2/9,5/9) | (4/9,2/9,4/9) | (4/9,2/9,5/9) | (4/9,2/9,7/9) | (4/9,2/9,1/3) |
| Mode | (4/9,2/9,5/9) | (4/9,2/9,7/9) | (4/9,2/9,5/9) | (4/9,2/9,5/9) | (4/9,2/9,5/9) | (4/9,2/9,2/3) | (4/9,2/9,4/9) | (4/9,2/9,5/9) | (4/9,2/9,1/3) |
| KNN | (4/9,2/9,5/9) | (4/9,2/9,1/3) | (4/9,2/9,4/9) | (4/9,2/9,5/9) | (4/9,2/9,2/3) | (4/9,2/9,1/3) | (4/9,2/9,5/9) | (4/9,2/9,4/9) | (4/9,2/9,7/9) |
| OT | (4/9,2/9,5/9) | (4/9,2/9,5/9) | (4/9,2/9,2/3) | (4/9,2/9,1/3) | (4/9,2/9,5/9) | (4/9,2/9,5/9) | (4/9,2/9,7/9) | (4/9,2/9,5/9) | (4/9,2/9,4/9) |
| random | (4/9,2/9,2/3) | (4/9,2/9,7/9) | (4/9,2/9,2/3) | (4/9,2/9,4/9) | (4/9,2/9,7/9) | (4/9,2/9,2/3) | (4/9,2/9,2/3) | (4/9,2/9,7/9) | (4/9,2/9,2/3) |
| MICE | (4/9,2/9,5/9) | (4/9,2/9,7/9) | (4/9,2/9,5/9) | (4/9,2/9,5/9) | (4/9,2/9,1/3) | (4/9,2/9,4/9) | (4/9,2/9,2/3) | (4/9,2/9,1/3) | (4/9,2/9,5/9) |
| Interpolation | (4/9,2/9,5/9) | (5/9,2/9,1/3) | (4/9,2/9,2/3) | (4/9,2/9,4/9) | (4/9,2/9,4/9) | (4/9,2/9,4/9) | (4/9,2/9,2/3) | (4/9,2/9,5/9) | (4/9,2/9,2/3) |
| Random Forest | (4/9,2/9,5/9) | (4/9,2/9,5/9) | (4/9,2/9,2/3) | (4/9,2/9,2/3) | (4/9,2/9,2/3) | (4/9,2/9,2/3) | (4/9,2/9,2/3) | (4/9,2/9,4/9) | (4/9,2/9,7/9) |
| LRR | (4/9,2/9,2/3) | (4/9,2/9,4/9) | (4/9,2/9,4/9) | (4/9,2/9,7/9) | (4/9,2/9,5/9) | (4/9,2/9,1/3) | (4/9,2/9,2/3) | (4/9,2/9,4/9) | (4/9,2/9,5/9) |
| MLP RR | (4/9,2/9,5/9) | (4/9,2/9,5/9) | (4/9,2/9,5/9) | (4/9,2/9,5/9) | (4/9,2/9,2/3) | (4/9,2/9,2/3) | (4/9,2/9,2/3) | (4/9,2/9,5/9) | (4/9,2/9,4/9) |

(b) `Condition-2B`$_j^{tech}$ across 3 data valuation methods, 12 imputation methods, and 9 missingness conditions; this measures the fraction of datasets for which the data subsampling protocol results in a class balance value less than the original class balance of the unsampled dataset (see Appendix Section D). Each cell value denotes a triplet of results for the three data valuation techniques: (`Condition-2B`$_j^{TMC-Shapley}$, `Condition-2B`$_j^{G-Shapley}$, `Condition-2B`$_j^{LOO}$), where $j$ denotes the imputation algorithm used on the data. Results are specific to experimental conditions in which the subsampled data is 80% of **lowest value data**. The highlighted values in teal color denote settings where subsampled data has a class balance greater than the full "unsampled" data, for the majority of the datasets. Excluding high-value data via TMC-Shapley and G-Shapley generally results in class balance greater than the unsampled set.

Table 4: `Condition-2A`$_j^{tech}$ and `Condition-2B`$_j^{tech}$ on datasets subsampled by selecting the lowest value data (80%). Generally, class balance improves as the result of subsampling, both (**a**) in overall class balance scores ($b$ greater than 0.25), and (**b**) relatively with respect to the unsampled dataset.

## G    POTENTIAL ADVERSE EFFECTS OF FAIRNESS AND GROUP REPRESENTATION

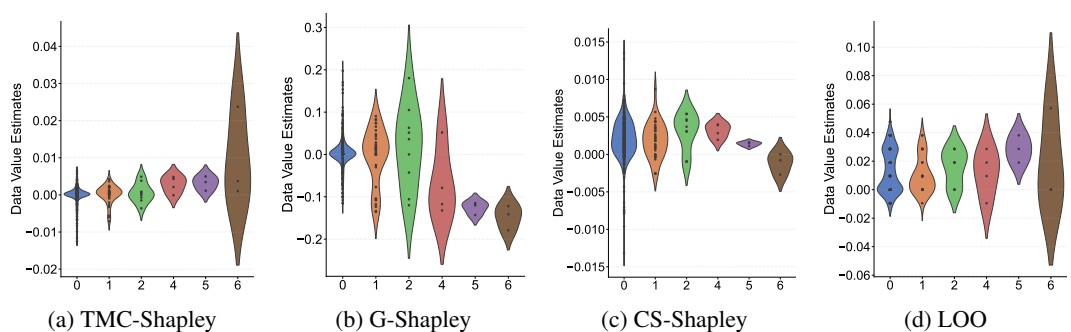

(a) TMC-Shapley        (b) G-Shapley        (c) CS-Shapley        (d) LOO

Figure 13: Data value distributions for dataset 1063 (KC2 Software defect prediction, random, MNAR-30) according to attribute group ("locodeandcomment").

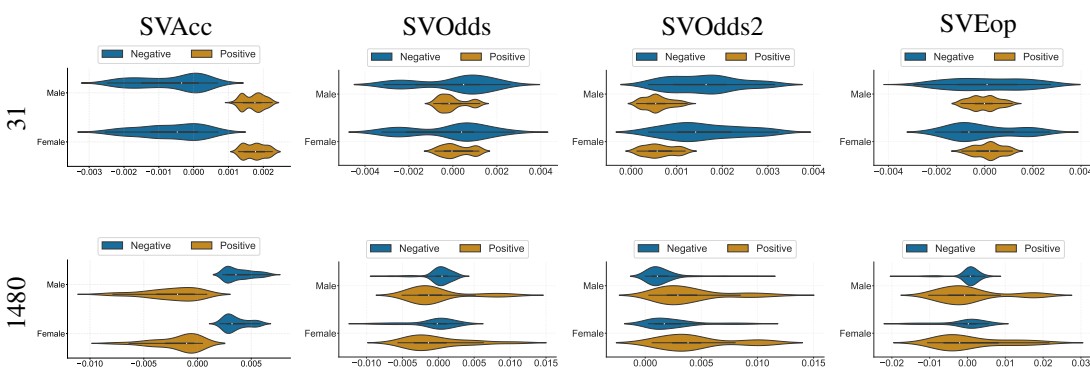

Figure 14: Distributions of accuracy and fairness Shapley values computed with FairShap on datasets 31 (German credit) and 1480 (Indian liver patient) with row removal and MCAR:30.

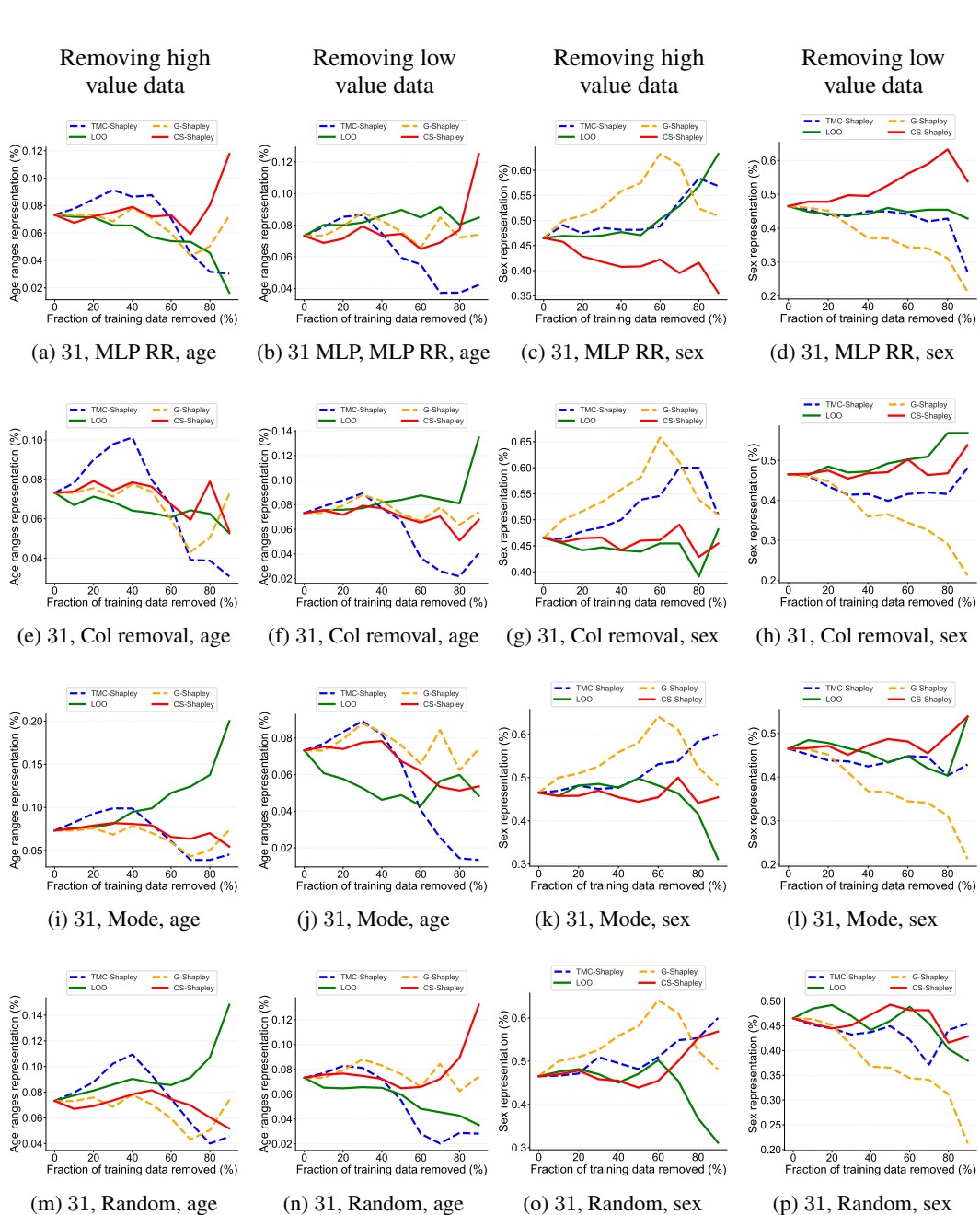

Figure 15: Percentage attribute representation (as defined in Appendix D as $g$, for binary sex and age) versus percentage of data removed, as a function of four data valuation metrics (TMC-Shapley, G-Shapley, LOO and CS-Shapley). Subfigure captions report the dataset label, imputation method, and (sensitive) attribute. All examples shown here have missingness pattern/percentage MNAR-30. The impact on group representation varies as a function of imputation method, valuation scheme, and removal of high- or low-valued data.

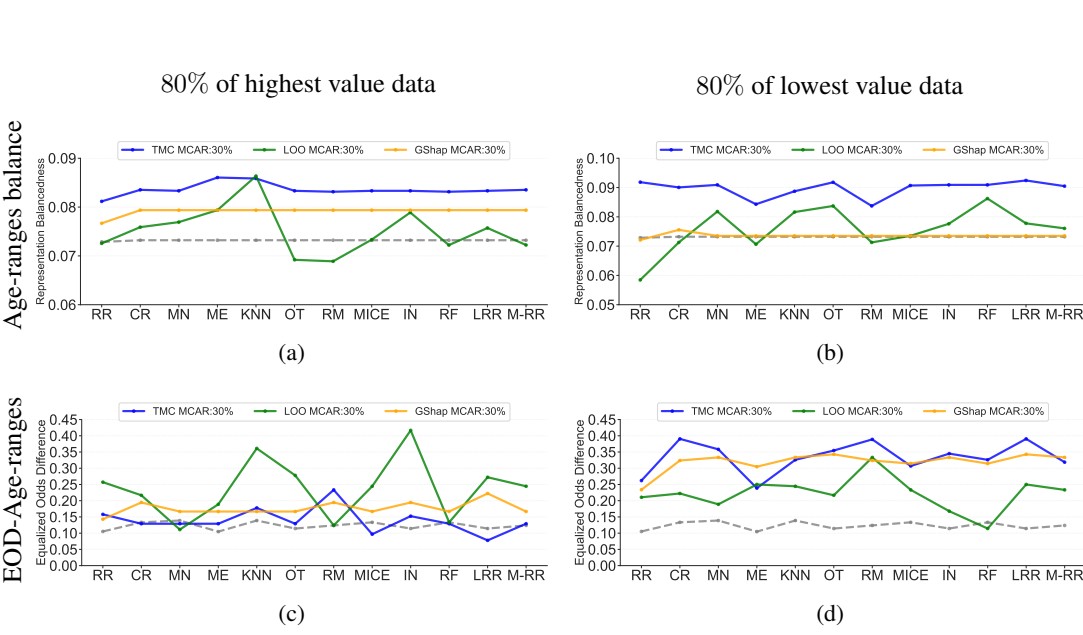

Figure 16: Representation balance (as defined in Appendix D as $g$) and equalized-odds difference (EOD) as a function of imputation algorithm and data valuation method (TMC-Shapley, LOO, and G-Shapley). Results are shown for dataset 1480 (Indian liver patient) and the attribute "age range". Abbreviations in the x-axis correspond to the imputation algorithm: ['Row Removal', 'Column Removal', 'Mean', 'Mode', 'KNN', 'OT', 'random', 'MICE', 'Interpolation', 'Random Forest', 'LRR', 'MLP RR']. The grey dotted line denotes the representation value when all the data (no sub-sampling) is used. Regardless of which data imputation algorithm used, the representation balance is higher than the unsampled data score when data is sampled via TMC-Shapley. G-Shapley results in a similar effect when low-value data is excluded. EOD tends to increase when low-valued data is excluded.

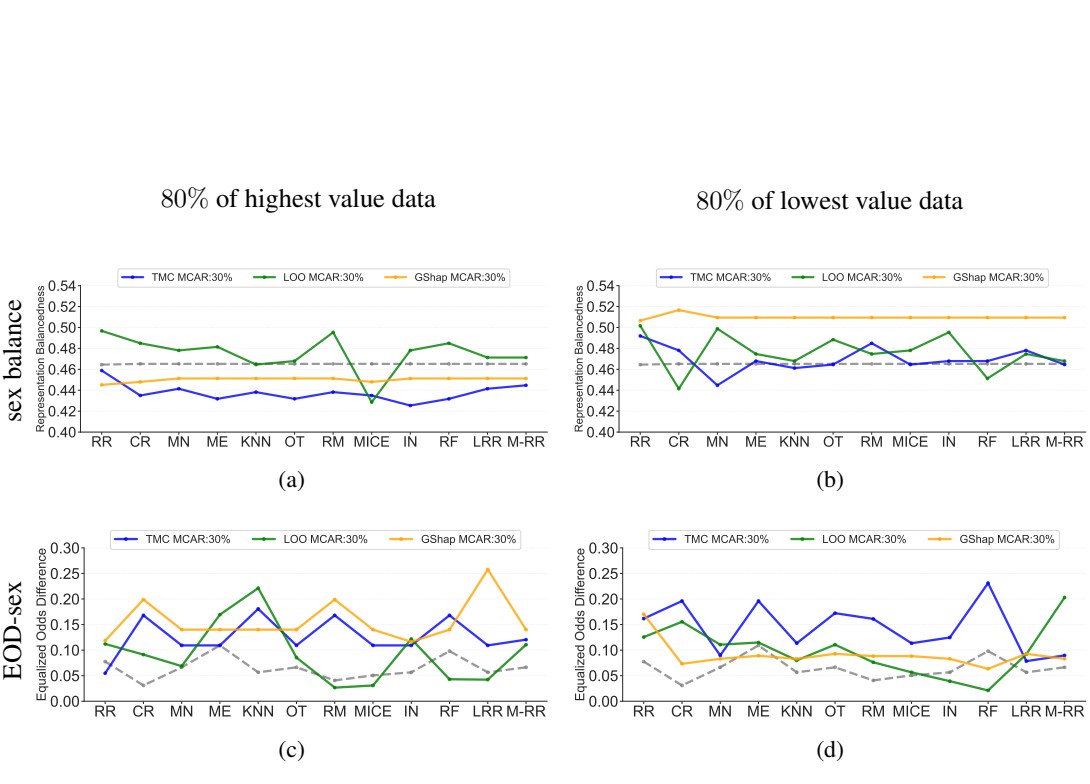

Figure 17: Representation balance (as defined in Appendix D as $g$) and equalized-odds difference (EOD) as a function of imputation algorithm and data valuation method (TMC-Shapley, LOO, and G-Shapley). Results are shown for dataset 31 (German credit) and the attribute "sex". Abbreviations in the x-axis correspond to the imputation algorithm: ['Row Removal', 'Column Removal', 'Mean', 'Mode', 'KNN', 'OT', 'random', 'MICE', 'Interpolation', 'Random Forest', 'LRR', 'MLP RR']. The grey dotted line denotes the representation value when all the data (no subsampling) is used. Regardless of which data imputation algorithm used, the representation balance is lower than the unsampled data score when data is sampled via TMC-Shapley and low-value data is excluded. G-Shapley results in lower or higher balance depending on whether high- or low-valued data is excluded, respectively. Most subsampling conditions result in increased EOD.

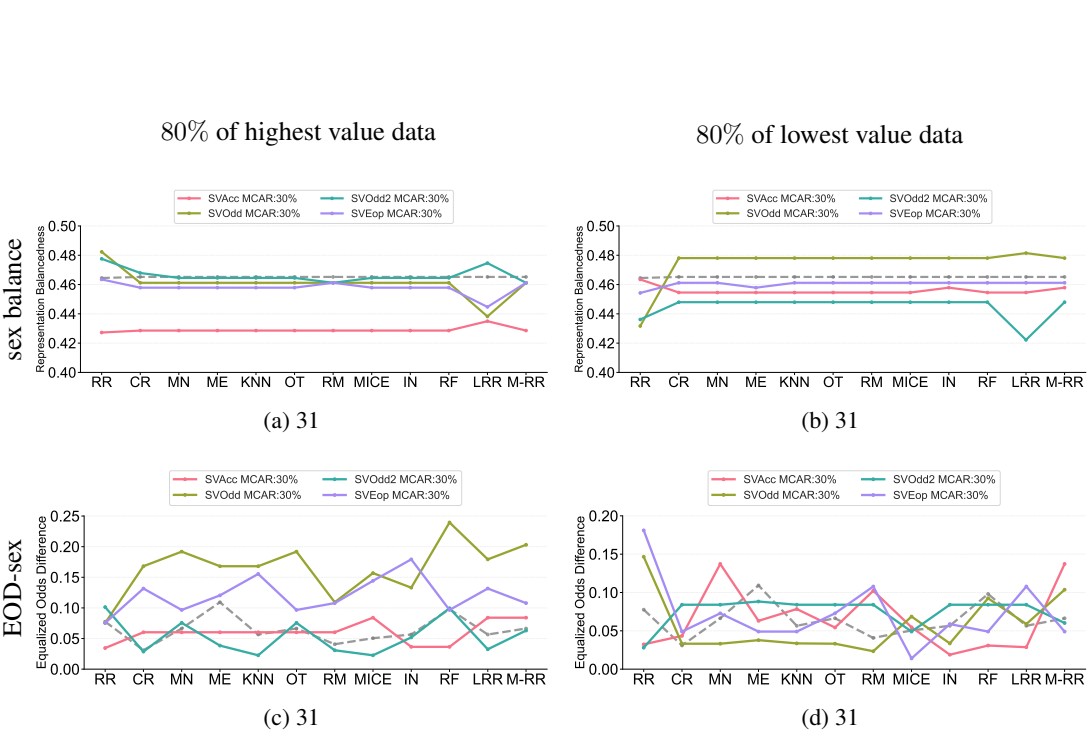

Figure 18: Representation balance (as defined in Appendix D as $g$) and equalized-odds difference (EOD) as a function of imputation algorithm and accuracy-/fairness-based data valuation method from FairShap (SVAcc, SVOdd, SVOdd2, SVEOP). Results are shown for dataset 31 (German credit) and the attribute "sex". Abbreviations in the x-axis correspond to the imputation algorithm: ['Row Removal', 'Column Removal', 'Mean', 'Mode', 'KNN', 'OT', 'random', 'MICE', 'Interpolation', 'Random Forest', 'LRR', 'MLP RR']. The grey dotted line denotes the representation value when all the data (no subsampling) is used. For most data imputation algorithms used, the representation balance is lower than the unsampled data score when data is sampled via SVAcc. Greater variance is observed in EOD.

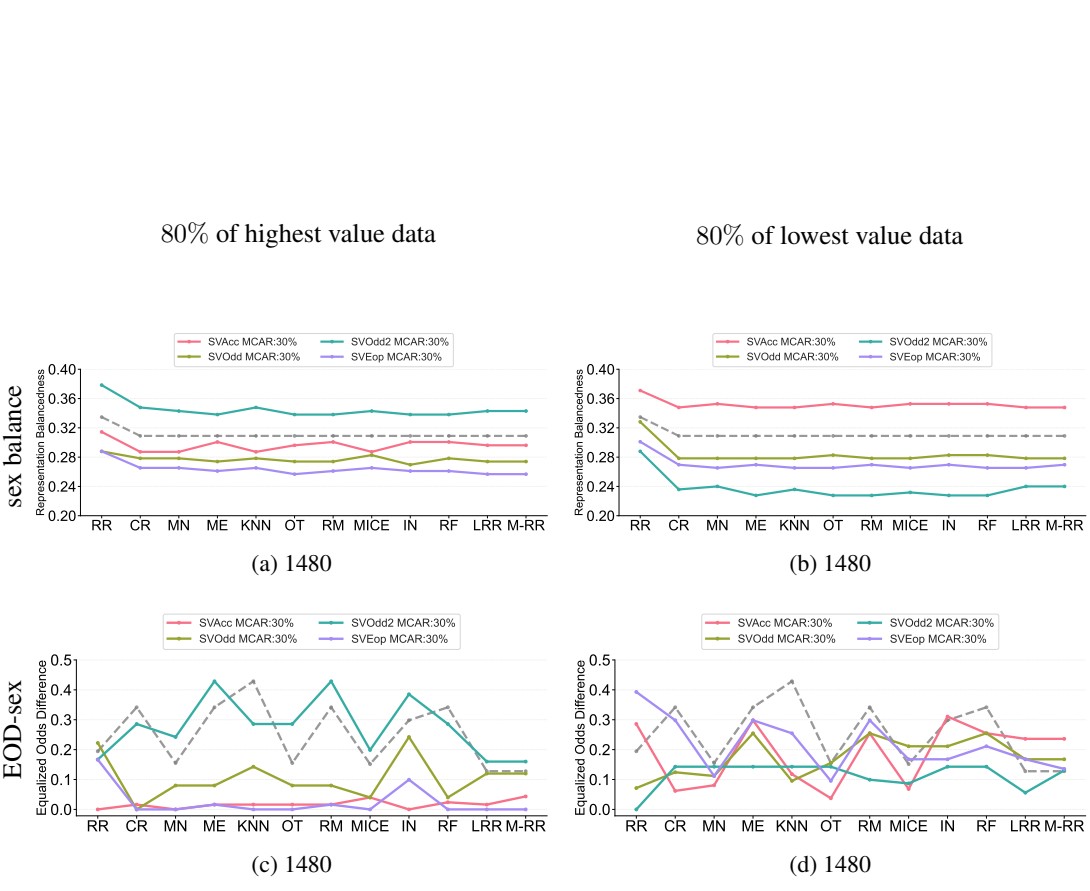

(a) 1480

(b) 1480

(c) 1480

(d) 1480

Figure 19: Representation balance (as defined in Appendix D as $g$) and equalized-odds difference (EOD) as a function of imputation algorithm and accuracy-/fairness-based data valuation method from FairShap (SVAcc, SVOdd, SVOdd2, SVEOP). Results are shown for dataset 1480 (Indian liver patient) and the attribute "sex". Abbreviations in the x-axis correspond to the imputation algorithm: ['Row Removal', 'Column Removal', 'Mean', 'Mode', 'KNN', 'OT', 'random', 'MICE', 'Interpolation', 'Random Forest', 'LRR', 'MLP RR']. The grey dotted line denotes the representation value when all the data (no subsampling) is used. For all data imputation algorithms used, the representation balance is lower than the unsampled data score when data is sampled via SVAcc, SVOdd, and SVEop and low-valued data is excluded. Likewise, the representation balance is typically lower than the unsampled data score when data is sampled via SVOdd, SVOdd2 and SVEop and high-valued data is excluded. Greater variance is observed in EOD; when low-value data is excluded, SVOdd2 tracks similarly to the unsampled data results, with other valuation methods resulting in lower EOD.

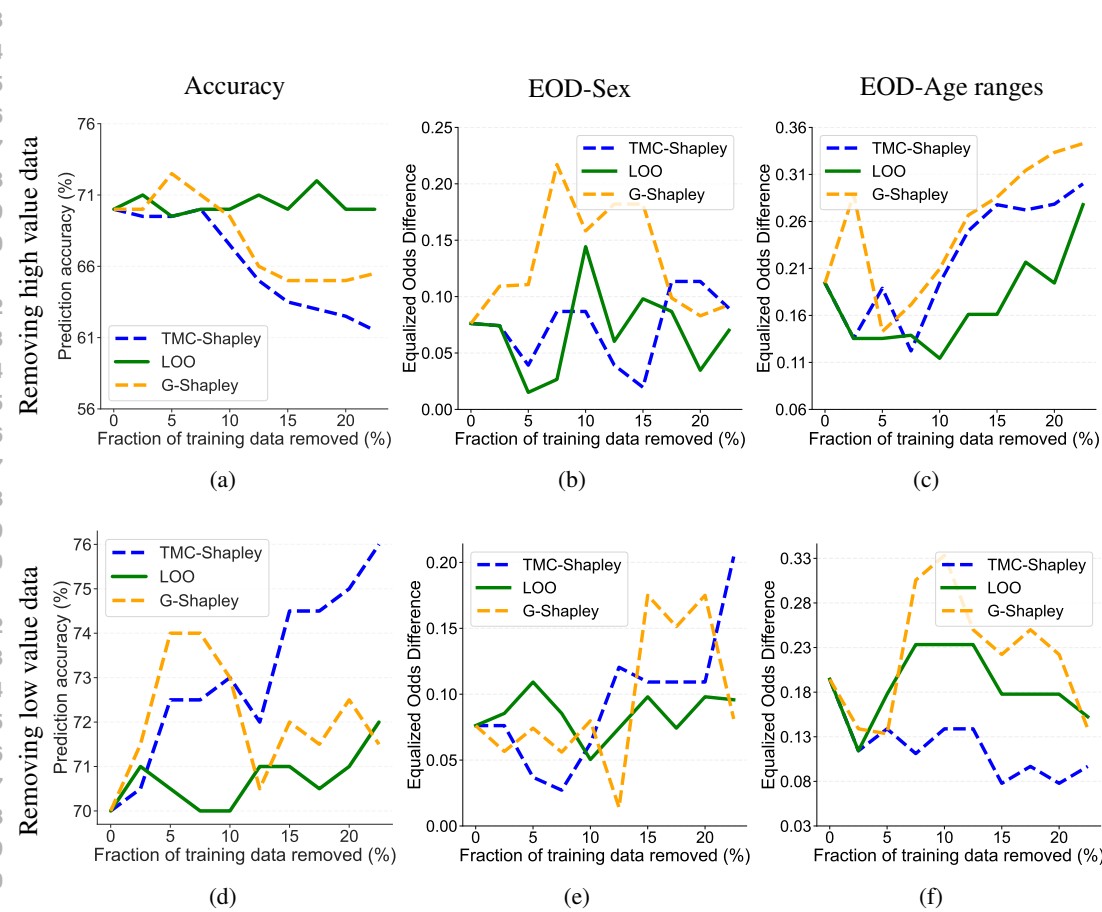

Figure 20: (**a**) Prediction accuracy and equalized-odds difference for (**b**) binary sex and (**c**) age range as a function of subsampling fraction for dataset 31 (German credit, MAR-30) and three data valuation methods (TMC-Shapley, LOO and G-Shapley). Generally, the removal of high-valued data decreases accuracy, with TMC-Shapley resulting in the greatest decrease. Removal of low-valued data shows the opposite trend. EOD tends to increase in both cases.

| | MAR:1 | MAR:10 | MAR:30 | MNAR:1 | MNAR:10 | MNAR:30 | MCAR:1 | MCAR:10 | MCAR:30 |
|---|---|---|---|---|---|---|---|---|---|
| Row Removal | (0,0,0) | (0,0,0) | (1,1,1) | (0,0,0) | (1,1,1) | (0,1,0) | (0,0,1) | (0,0,0) | (1,0,0) |
| Col Removal | (0,0,0) | (0,0,0) | (0,0,0) | (0,0,0) | (0,0,0) | (0,0,0) | (0,0,0) | (0,0,0) | (0,0,0) |
| Mean | (0,0,0) | (0,0,1) | (0,0,0) | (0,0,0) | (0,0,1) | (0,0,1) | (0,0,1) | (0,0,0) | (0,0,0) |
| Mode | (0,0,0) | (0,0,0) | (0,0,0) | (0,0,0) | (0,0,1) | (0,0,0) | (0,0,1) | (0,0,1) | (0,0,0) |
| KNN | (0,0,0) | (0,0,0) | (0,0,1) | (0,0,0) | (0,0,1) | (1,0,0) | (0,0,1) | (0,0,0) | (0,0,0) |
| OT | (0,0,0) | (0,0,1) | (0,0,0) | (0,0,0) | (0,0,0) | (1,0,1) | (0,0,0) | (0,0,1) | (0,0,0) |
| random | (0,0,0) | (0,0,0) | (0,0,0) | (0,0,0) | (0,0,1) | (0,0,1) | (0,0,0) | (0,0,0) | (0,0,1) |
| MICE | (0,0,0) | (0,0,0) | (0,0,0) | (0,0,0) | (0,0,0) | (0,0,0) | (0,0,0) | (0,0,0) | (0,0,1) |
| Interpolation | (0,0,1) | (0,0,0) | (0,0,0) | (0,0,0) | (1,0,1) | (1,0,0) | (0,0,0) | (0,1,0) | (0,0,0) |
| Random Forest | (0,0,0) | (0,0,0) | (0,0,0) | (0,0,0) | (1,1,1) | (0,0,0) | (0,0,1) | (0,0,1) | (0,0,1) |
| LRR | (0,0,0) | (0,1,0) | (0,0,0) | (0,0,0) | (0,0,0) | (0,0,1) | (0,0,0) | (0,0,0) | (0,0,1) |
| MLP RR | (0,0,0) | (0,0,1) | (0,0,1) | (0,0,0) | (1,0,1) | (0,0,1) | (0,0,1) | (0,0,1) | (0,0,0) |

(a) $\texttt{Condition-3}_j^{tech}$ across 3 data valuation methods, 12 imputation methods, and 9 missingness conditions; this measures whether or not the data subsampling protocol results in an EOD value less than the original EOD of the unsampled dataset; i.e., is 1 if it is "more fair" (see Appendix Section D). Each cell value denotes a triplet of results for the three data valuation techniques: ($\texttt{Condition-3}_j^{TMC-Shapley}$, $\texttt{Condition-3}_j^{G-Shapley}$, $\texttt{Condition-3}_j^{LOO}$), where $j$ is the imputation algorithm. Results are specific to experimental conditions in which the subsampled data is $80\%$ of **highest value data** and the sensitive group is *sex*. The highlighted triplets denote cases where subsampling improves fairness via all three data valuation techniques.

| | MAR:1 | MAR:10 | MAR:30 | MNAR:1 | MNAR:10 | MNAR:30 | MCAR:1 | MCAR:10 | MCAR:30 |
|---|---|---|---|---|---|---|---|---|---|
| Row Removal | (0,0,1) | (0,1,1) | (0,1,1) | (0,0,0) | (1,1,0) | (1,0,1) | (0,0,0) | (0,0,0) | (0,0,0) |
| Col Removal | (0,0,0) | (0,0,0) | (0,0,0) | (0,0,0) | (0,0,0) | (0,0,0) | (0,0,0) | (0,0,0) | (0,0,0) |
| Mean | (0,0,0) | (0,0,0) | (0,0,1) | (0,0,0) | (0,0,0) | (1,0,0) | (0,0,0) | (0,0,1) | (0,0,0) |
| Mode | (0,0,1) | (0,0,0) | (0,0,1) | (0,0,1) | (0,0,0) | (1,0,0) | (0,0,0) | (0,0,1) | (0,1,0) |
| KNN | (0,0,0) | (0,0,0) | (0,1,0) | (0,0,0) | (0,0,0) | (1,1,1) | (0,0,0) | (0,1,0) | (0,0,0) |
| OT | (0,0,0) | (0,0,1) | (0,0,0) | (0,0,0) | (0,0,0) | (1,0,0) | (0,0,1) | (0,0,0) | (0,0,0) |
| random | (0,0,1) | (0,0,0) | (0,0,1) | (0,0,0) | (0,1,0) | (0,0,0) | (0,0,1) | (0,0,0) | (0,0,0) |
| MICE | (0,0,0) | (0,0,0) | (0,0,0) | (0,0,1) | (0,0,0) | (0,0,1) | (0,0,0) | (0,0,0) | (0,0,0) |
| Interpolation | (0,0,1) | (0,0,0) | (0,0,0) | (0,0,0) | (0,0,0) | (1,0,1) | (0,0,1) | (0,1,0) | (0,0,1) |
| Random Forest | (0,0,1) | (0,0,0) | (0,1,0) | (0,0,0) | (1,1,0) | (1,1,0) | (0,0,0) | (0,0,0) | (0,1,1) |
| LRR | (0,0,0) | (0,0,0) | (0,1,0) | (0,0,0) | (0,0,0) | (0,0,0) | (0,0,1) | (0,0,0) | (0,0,0) |
| MLP RR | (0,0,0) | (0,0,0) | (0,0,0) | (0,0,0) | (1,0,1) | (0,0,0) | (0,0,0) | (0,0,0) | (0,0,0) |

(b) $\texttt{Condition-3}_j^{tech}$ across 3 data valuation methods, 12 imputation methods, and 9 missingness conditions; this measures whether or not the data subsampling protocol results in an EOD value less than the original EOD of the unsampled dataset; i.e., is 1 if it is "more fair" (see Appendix Section D). Each cell value denotes a triplet of results for the three data valuation techniques: ($\texttt{Condition-3}_j^{TMC-Shapley}$, $\texttt{Condition-3}_j^{G-Shapley}$, $\texttt{Condition-3}_j^{LOO}$), where $j$ is the imputation algorithm. Results are specific to experimental conditions in which the subsampled data is $80\%$ of **lowest value data** and the sensitive group is *sex*. The highlighted triplets denote cases where subsampling improves fairness via all three data valuation techniques.

Table 5: $\texttt{Condition-3}_j^{tech}$ with attribute *sex* on datasets subsampled by selecting the (**a**) highest and (**b**) lowest value data ($80\%$). In both cases, fairness generally worsens as the result of subsampling.

| | MAR:1 | MAR:10 | MAR:30 | MNAR:1 | MNAR:10 | MNAR:30 | MCAR:1 | MCAR:10 | MCAR:30 |
|---|---|---|---|---|---|---|---|---|---|
| Row Removal | (1,0,0) | (0,0,1) | (1,1,1) | (1,0,0) | (1,1,1) | (0,0,0) | (1,0,0) | (0,1,0) | (0,0,0) |
| Col Removal | (1,0,0) | (1,0,0) | (1,0,0) | (1,0,0) | (1,0,0) | (1,0,0) | (1,0,0) | (1,0,0) | (1,0,0) |
| Mean | (1,0,1) | (0,0,0) | (1,0,1) | (1,0,1) | (1,0,0) | (0,0,0) | (1,0,0) | (0,0,0) | (1,0,1) |
| Mode | (1,0,0) | (1,0,0) | (0,0,0) | (1,0,1) | (1,0,1) | (0,0,0) | (1,0,0) | (1,0,0) | (0,0,0) |
| KNN | (1,0,1) | (0,0,0) | (1,0,0) | (1,0,1) | (1,0,0) | (1,0,0) | (1,0,1) | (0,0,0) | (0,0,0) |
| OT | (1,0,0) | (0,0,0) | (1,0,0) | (1,0,1) | (1,0,0) | (1,0,0) | (1,0,0) | (0,0,0) | (0,0,0) |
| random | (1,0,0) | (1,0,0) | (0,0,0) | (1,0,1) | (1,0,0) | (1,0,0) | (1,0,1) | (1,1,1) | (0,0,0) |
| MICE | (1,0,1) | (1,0,0) | (0,0,0) | (1,0,1) | (1,0,0) | (1,0,1) | (1,0,0) | (0,0,0) | (1,0,0) |
| Interpolation | (1,0,0) | (0,0,0) | (1,0,0) | (1,0,1) | (1,0,0) | (0,0,0) | (1,0,0) | (0,0,0) | (0,0,0) |
| Random Forest | (1,0,1) | (1,0,0) | (1,0,0) | (1,0,1) | (1,0,0) | (1,0,1) | (1,0,0) | (0,0,0) | (1,0,0) |
| LRR | (0,0,0) | (1,0,0) | (0,0,0) | (1,0,0) | (0,0,0) | (0,0,0) | (1,0,1) | (1,0,0) | (1,0,0) |
| MLP RR | (0,0,0) | (0,0,0) | (1,0,0) | (1,0,1) | (0,0,0) | (0,0,0) | (1,0,0) | (0,0,0) | (0,0,0) |

(a) Condition-$3_j^{tech}$ across 3 data valuation methods, 12 imputation methods, and 9 missingness conditions; this measures whether or not the data subsampling protocol results in an EOD value less than the original EOD of the unsampled dataset; i.e., is 1 if it is "more fair" (see Appendix Section D). Each cell value denotes a triplet of results for the three data valuation techniques: (Condition-$3_j^{TMC-Shapley}$, Condition-$3_j^{G-Shapley}$, Condition-$3_j^{LOO}$), where $j$ is the imputation algorithm. Results are specific to experimental conditions in which the subsampled data is $80\%$ of **highest value data** and the sensitive group is *age range*. The highlighted triplets denote cases where subsampling improves fairness via all three data valuation techniques.

| | MAR:1 | MAR:10 | MAR:30 | MNAR:1 | MNAR:10 | MNAR:30 | MCAR:1 | MCAR:10 | MCAR:30 |
|---|---|---|---|---|---|---|---|---|---|
| Row Removal | (0,0,0) | (0,0,0) | (0,1,1) | (0,0,0) | (0,0,0) | (0,0,0) | (0,0,1) | (1,0,1) | (0,0,0) |
| Col Removal | (0,0,0) | (0,0,0) | (0,0,0) | (0,0,0) | (0,0,0) | (0,0,0) | (0,0,0) | (0,0,0) | (0,0,0) |
| Mean | (0,0,0) | (0,0,0) | (0,0,0) | (0,0,0) | (0,0,1) | (0,0,0) | (0,0,0) | (0,0,0) | (0,0,0) |
| Mode | (0,0,0) | (0,0,0) | (0,0,0) | (0,0,0) | (0,0,0) | (0,0,0) | (0,0,0) | (0,0,0) | (0,0,0) |
| KNN | (0,0,1) | (0,0,1) | (0,0,0) | (0,0,0) | (0,0,0) | (0,0,0) | (0,0,0) | (0,0,0) | (0,0,0) |
| OT | (0,0,1) | (0,0,0) | (0,0,0) | (0,0,1) | (0,0,1) | (0,0,0) | (0,0,1) | (0,0,0) | (0,0,0) |
| random | (0,0,0) | (0,0,0) | (0,0,0) | (0,0,1) | (0,0,0) | (0,0,0) | (0,0,0) | (0,0,1) | (0,0,0) |
| MICE | (0,0,1) | (0,0,1) | (0,0,1) | (0,0,1) | (0,0,0) | (0,0,1) | (0,0,0) | (0,0,0) | (0,0,0) |
| Interpolation | (0,0,0) | (0,0,0) | (0,0,0) | (0,0,1) | (0,0,0) | (0,0,0) | (0,0,1) | (0,0,0) | (0,0,0) |
| Random Forest | (0,0,0) | (0,0,1) | (0,0,0) | (0,0,0) | (0,0,0) | (0,0,0) | (0,0,0) | (0,0,1) | (0,0,1) |
| LRR | (0,0,0) | (0,0,0) | (0,0,0) | (0,0,0) | (0,0,1) | (0,0,0) | (0,0,0) | (0,0,0) | (0,0,0) |
| MLP RR | (0,0,0) | (0,0,1) | (0,0,0) | (0,0,1) | (0,0,1) | (0,0,0) | (0,0,0) | (0,0,0) | (0,0,0) |

(b) Condition-$3_j^{tech}$ across 3 data valuation methods, 12 imputation methods, and 9 missingness conditions; this measures whether or not the data subsampling protocol results in an EOD value less than the original EOD of the unsampled dataset; i.e., is 1 if it is "more fair" (see Appendix Section D). Each cell value denotes a triplet of results for the three data valuation techniques: (Condition-$3_j^{TMC-Shapley}$, Condition-$3_j^{G-Shapley}$, Condition-$3_j^{LOO}$), where $j$ is the imputation algorithm. Results are specific to experimental conditions in which the subsampled data is $80\%$ of **lowest value data** and the sensitive group is *age range*. The highlighted triplets denote cases where subsampling improves fairness via all three data valuation techniques.

Table 6: Condition-$3_j^{tech}$ with attribute *age range* on datasets subsampled by selecting the (**a**) highest and (**b**) lowest value data ($80\%$). In both cases, fairness generally worsens as the result of subsampling.

|  | MAR:1 | MAR:10 | MAR:30 | MNAR:1 | MNAR:10 | MNAR:30 | MCAR:1 | MCAR:10 | MCAR:30 |
|---|---|---|---|---|---|---|---|---|---|
| Row Removal | (1,1,1) | (1,1,0) | (1,1,1) | (1,1,0) | (1,1,1) | (1,1,0) | (1,1,0) | (1,1,1) | (1,1,0) |
| Col Removal | (1,1,0) | (1,1,0) | (1,1,0) | (1,1,0) | (1,1,0) | (1,1,0) | (1,1,0) | (1,1,0) | (1,1,0) |
| Mean | (1,1,0) | (1,1,0) | (1,1,0) | (1,1,1) | (1,1,1) | (1,1,0) | (1,1,0) | (1,1,1) | (1,1,0) |
| Mode | (1,1,1) | (1,1,1) | (1,1,1) | (1,1,0) | (1,1,0) | (1,1,0) | (1,1,0) | (1,1,1) | (1,1,0) |
| KNN | (1,1,1) | (1,1,0) | (1,1,1) | (1,1,0) | (1,1,1) | (1,1,1) | (1,1,0) | (1,1,1) | (1,1,1) |
| OT | (1,1,0) | (1,1,0) | (1,1,1) | (1,1,0) | (1,1,1) | (1,1,1) | (1,1,0) | (1,1,0) | (1,1,0) |
| random | (1,1,0) | (1,1,1) | (1,1,1) | (1,1,0) | (1,1,1) | (1,1,0) | (1,1,0) | (1,1,1) | (1,1,0) |
| MICE | (1,1,1) | (1,1,0) | (1,1,1) | (1,1,0) | (1,1,1) | (1,1,1) | (1,1,1) | (1,1,1) | (1,1,1) |
| Interpolation | (1,1,0) | (1,1,0) | (1,1,0) | (1,1,0) | (1,1,1) | (1,1,0) | (1,1,0) | (1,1,0) | (1,1,0) |
| Random Forest | (1,1,1) | (1,1,0) | (1,1,0) | (1,1,0) | (1,1,0) | (1,1,0) | (1,1,0) | (1,1,0) | (1,1,0) |
| LRR | (1,1,1) | (1,1,0) | (1,1,0) | (1,1,0) | (1,1,0) | (1,1,0) | (1,1,1) | (1,1,1) | (1,1,0) |
| MLP RR | (1,1,1) | (1,1,1) | (1,1,1) | (1,1,1) | (1,1,0) | (1,1,1) | (1,1,0) | (1,1,0) | (1,1,0) |

(a) $\texttt{Condition-4}_j^{tech}$ across 3 data valuation methods, 12 imputation methods, and 9 missingness conditions; this measures whether or not the data subsampling protocol results in a group (or attribute) representation balance value less than the original balance value of the unsampled dataset; i.e., is 1 if it is "less balanced" (see Appendix Section D). Each cell value denotes a triplet of results for the three data valuation techniques: ($\texttt{Condition-4}_j^{TMC-Shapley}$, $\texttt{Condition-4}_j^{G-Shapley}$, $\texttt{Condition-4}_j^{LOO}$), where $j$ is the imputation algorithm. Results are specific to experimental conditions in which the subsampled data is $80\%$ of **highest value data** and the sensitive group is *sex*. The highlighted triplets are cases where subsampling improves the balance of the sensitive group regardless of the data valuation technique used for subsampling.

|  | MAR:1 | MAR:10 | MAR:30 | MNAR:1 | MNAR:10 | MNAR:30 | MCAR:1 | MCAR:10 | MCAR:30 |
|---|---|---|---|---|---|---|---|---|---|
| Row Removal | (0,0,1) | (0,0,0) | (1,0,1) | (1,0,1) | (1,0,0) | (0,0,0) | (0,0,1) | (1,0,0) | (0,0,0) |
| Col Removal | (0,0,1) | (0,0,1) | (0,0,1) | (0,0,1) | (0,0,1) | (0,0,1) | (0,0,1) | (0,0,1) | (0,0,1) |
| Mean | (0,0,1) | (1,0,0) | (0,0,0) | (0,0,1) | (1,0,0) | (0,0,0) | (0,0,0) | (0,0,0) | (1,0,0) |
| Mode | (0,0,0) | (1,0,0) | (1,0,0) | (1,0,0) | (1,0,1) | (0,0,0) | (0,0,0) | (1,0,0) | (0,0,0) |
| KNN | (0,0,1) | (1,0,1) | (0,0,1) | (0,0,1) | (0,0,0) | (0,0,0) | (1,0,1) | (1,0,0) | (1,0,0) |
| OT | (1,0,1) | (0,0,1) | (0,0,1) | (0,0,1) | (0,0,0) | (0,0,1) | (1,0,0) | (0,0,1) | (1,0,0) |
| random | (1,0,1) | (1,0,0) | (1,0,0) | (1,0,1) | (0,0,1) | (0,0,0) | (0,0,1) | (1,0,0) | (0,0,0) |
| MICE | (0,0,1) | (0,0,1) | (0,0,1) | (0,0,1) | (0,0,1) | (1,0,1) | (0,0,1) | (1,0,1) | (1,0,0) |
| Interpolation | (0,0,1) | (1,0,0) | (0,0,1) | (1,0,1) | (0,0,0) | (0,0,1) | (1,0,1) | (1,0,0) | (0,0,0) |
| Random Forest | (0,0,1) | (0,0,1) | (0,0,1) | (0,0,1) | (1,0,1) | (0,0,1) | (0,0,1) | (1,0,1) | (0,0,1) |
| LRR | (0,0,0) | (1,0,1) | (0,0,1) | (1,0,0) | (0,0,1) | (0,0,0) | (1,0,0) | (0,0,1) | (0,0,0) |
| MLP RR | (0,0,0) | (0,0,0) | (0,0,0) | (1,0,1) | (0,0,1) | (0,0,0) | (1,0,0) | (0,0,1) | (1,0,0) |

(b) $\texttt{Condition-4}_j^{tech}$ across 3 data valuation methods, 12 imputation methods, and 9 missingness conditions; this measures whether or not the data subsampling protocol results in a group (or attribute) representation balance value less than the original balance value of the unsampled dataset; i.e., is 1 if it is "less balanced" (see Appendix Section D). Each cell value denotes a triplet of results for the three data valuation techniques: ($\texttt{Condition-4}_j^{TMC-Shapley}$, $\texttt{Condition-4}_j^{G-Shapley}$, $\texttt{Condition-4}_j^{LOO}$), where $j$ is the imputation algorithm. Results are specific to experimental conditions in which the subsampled data is $80\%$ of **lowest value data** and the sensitive group is *sex*. The highlighted triplets are cases where subsampling improves the balance of the sensitive group regardless of the data valuation technique used for subsampling.

Table 7: $\texttt{Condition-4}_j^{tech}$ with attribute *sex* on datasets subsampled by selecting the (**a**) highest and (**b**) lowest value data ($80\%$). Representation balance generally worsens as the result of excluding low-valued data, whereas exclusion of high-valued data has more varied effects.

|               | MAR:1   | MAR:10  | MAR:30  | MNAR:1  | MNAR:10 | MNAR:30 | MCAR:1  | MCAR:10 | MCAR:30 |
|---------------|---------|---------|---------|---------|---------|---------|---------|---------|---------|
| Row Removal   | (0,0,1) | (0,0,0) | (0,0,0) | (0,0,1) | (0,0,1) | (0,0,0) | (0,0,0) | (0,0,1) | (0,0,1) |
| Col Removal   | (0,0,0) | (0,0,0) | (0,0,0) | (0,0,0) | (0,0,0) | (0,0,0) | (0,0,0) | (0,0,0) | (0,0,0) |
| Mean          | (0,0,0) | (0,0,1) | (0,0,0) | (0,0,0) | (0,0,0) | (0,0,1) | (0,0,0) | (0,0,0) | (0,0,0) |
| Mode          | (0,0,0) | (0,0,0) | (0,0,1) | (0,0,0) | (0,0,1) | (0,0,1) | (0,0,0) | (0,0,0) | (0,0,0) |
| KNN           | (0,0,0) | (0,0,1) | (0,0,0) | (0,0,1) | (0,0,1) | (0,0,0) | (0,0,0) | (0,0,0) | (0,0,0) |
| OT            | (0,0,0) | (0,0,0) | (0,0,0) | (0,0,0) | (0,0,1) | (0,0,0) | (0,0,0) | (0,0,1) | (0,0,1) |
| random        | (0,0,0) | (0,0,0) | (0,0,1) | (0,0,1) | (0,0,0) | (0,0,1) | (0,0,0) | (0,0,0) | (0,0,1) |
| MICE          | (0,0,0) | (0,0,0) | (0,0,0) | (0,0,0) | (0,0,1) | (0,0,0) | (0,0,0) | (0,0,0) | (0,0,0) |
| Interpolation | (0,0,0) | (0,0,1) | (0,0,1) | (0,0,0) | (0,0,0) | (0,0,0) | (0,0,0) | (0,0,1) | (0,0,0) |
| Random Forest | (0,0,1) | (0,0,0) | (0,0,1) | (0,0,0) | (0,0,0) | (0,0,1) | (0,0,0) | (0,0,0) | (0,0,1) |
| LRR           | (0,0,1) | (0,0,1) | (0,0,0) | (0,0,1) | (0,0,1) | (0,0,1) | (0,0,1) | (0,0,1) | (0,0,0) |
| MLP RR        | (0,0,1) | (0,0,0) | (0,0,0) | (0,0,0) | (0,0,1) | (0,0,0) | (0,0,0) | (0,0,0) | (0,0,1) |

(a) `Condition-4`$_j^{tech}$ across 3 data valuation methods, 12 imputation methods, and 9 missingness conditions; this measures whether or not the data subsampling protocol results in a group (or attribute) representation balance value less than the original balance value of the unsampled dataset; i.e., is 1 if it is "less balanced" (see Appendix Section D). Each cell value denotes a triplet of results for the three data valuation techniques: (`Condition-4`$_j^{TMC-Shapley}$, `Condition-4`$_j^{G-Shapley}$, `Condition-4`$_j^{LOO}$), where $j$ is the imputation algorithm. Results are specific to experimental conditions in which the subsampled data is $80\%$ of **highest value data** and the sensitive group is *age range*. The highlighted triplets are cases where subsampling improves the balance of the sensitive group regardless of the data valuation technique used for subsampling.

|               | MAR:1   | MAR:10  | MAR:30  | MNAR:1  | MNAR:10 | MNAR:30 | MCAR:1  | MCAR:10 | MCAR:30 |
|---------------|---------|---------|---------|---------|---------|---------|---------|---------|---------|
| Row Removal   | (0,0,1) | (0,1,1) | (0,1,1) | (0,1,1) | (0,1,0) | (0,0,1) | (0,1,1) | (0,1,0) | (0,1,1) |
| Col Removal   | (0,0,1) | (0,0,1) | (0,0,1) | (0,0,1) | (0,0,1) | (0,0,1) | (0,0,1) | (0,0,1) | (0,0,1) |
| Mean          | (0,0,1) | (0,0,1) | (0,0,1) | (0,0,1) | (0,0,1) | (0,0,0) | (0,0,1) | (0,0,0) | (0,0,0) |
| Mode          | (0,0,1) | (0,0,0) | (0,0,0) | (0,0,1) | (0,0,0) | (0,0,0) | (0,0,1) | (0,0,1) | (0,0,1) |
| KNN           | (0,0,1) | (0,0,0) | (0,0,1) | (0,0,1) | (0,0,0) | (0,0,1) | (0,0,1) | (0,0,1) | (0,0,0) |
| OT            | (0,0,1) | (0,0,0) | (0,0,1) | (0,0,1) | (0,0,0) | (0,0,1) | (0,0,1) | (0,0,0) | (0,0,0) |
| random        | (0,0,0) | (0,0,0) | (0,0,0) | (0,0,1) | (0,0,1) | (0,0,0) | (0,0,0) | (0,0,0) | (0,0,1) |
| MICE          | (0,0,1) | (0,0,1) | (0,0,0) | (0,0,1) | (0,0,0) | (0,0,1) | (0,0,1) | (0,0,0) | (0,0,0) |
| Interpolation | (0,0,1) | (0,0,0) | (0,0,1) | (0,0,1) | (0,0,1) | (0,0,1) | (0,0,0) | (0,0,1) | (0,0,0) |
| Random Forest | (0,0,1) | (0,0,1) | (0,0,0) | (0,0,0) | (0,0,0) | (0,0,1) | (0,0,1) | (0,0,0) | (0,0,0) |
| LRR           | (0,0,1) | (0,0,1) | (0,0,1) | (0,0,0) | (0,0,0) | (0,0,0) | (0,0,0) | (0,0,0) | (0,0,0) |
| MLP RR        | (0,0,1) | (0,0,0) | (0,0,1) | (0,0,1) | (0,0,1) | (0,0,1) | (0,0,0) | (0,0,1) | (0,0,0) |

(b) `Condition-4`$_j^{tech}$ across 3 data valuation methods, 12 imputation methods, and 9 missingness conditions; this measures whether or not the data subsampling protocol results in a group (or attribute) representation balance value less than the original balance value of the unsampled dataset; i.e., is 1 if it is "less balanced" (see Appendix Section D). Each cell value denotes a triplet of results for the three data valuation techniques: (`Condition-4`$_j^{TMC-Shapley}$, `Condition-4`$_j^{G-Shapley}$, `Condition-4`$_j^{LOO}$), where $j$ is the imputation algorithm. Results are specific to experimental conditions in which the subsampled data is $80\%$ of **lowest value data** and the sensitive group is *age range*. The highlighted triplets are cases where subsampling improves the balance of the sensitive group regardless of the data valuation technique used for subsampling.

Table 8: `Condition-4`$_j^{tech}$ with attribute *age range* on datasets subsampled by selecting the (**a**) highest and (**b**) lowest value data ($80\%$). Representation balance generally improves as the result of subsampling for this attribute.

## H DVALCARD SECTION DESCRIPTIONS

The proposed DValCard framework consists of six sections: "Introduction", "System Flowchart", "DVal Candidate Data", "DVal Method", "DVal Report", and "Ethical Statement and Recommendations". We briefly describe the suggested section contents below.

---

### DValCard Template

Introduction providing details on the DValCard developer, including the date of its development and contact details of the developers.

**System Flowchart**
- DVal in the life cycle context

**DVal Candidate Data**
- Data information (datasheet)
- Data preprocessing

**DVal Method**
- DVal technique(s)
- Learning algorithm(s)
- Performance metric(s)
- Evaluation data

**DVal Report**

- Data values (describe)
- Excluded/removed instances (describe)
- Included/chosen instances (describe)

**Ethical Statement and Recommendations**

- Intended users, and in/out-of-scope use cases
- Potential ethical issues to consider
- Legal considerations
- Environmental considerations
- Recommendations

---

Figure 21: Proposed structure of a DValCard for data valuation transparency.

### H.1 INTRODUCTION

The DValCard introduction includes general details about the DValCard and its developers, including the date, the version of the card's development, and contact information for its authors, including at least one corresponding author.

### H.2 SYSTEM FLOWCHART

The system flowchart consists of a diagram detailing the complete life cycle of the data valuation method, with the purpose of illustrating where data valuation occurs with respect to other algorithmic design choices. Depending on the use case, data valuation may be part of data preprocessing, cleaning, or curation; or it may be conducted independently at the end of the data life cycle. Inclusion of a system flowchart promotes greater clarity in data value interpretation and usage. The key subsections of this section is:

**DVal in the life cycle context.** A pictoral flowchart indicating data valuation with respect to other processes in the system.

### H.3 DVAL CANDIDATE DATA

The phrase "DVal candidate data" pertains to the data to be processed to obtain the corresponding data values. The DVal candidate data can come from various sources. It is crucial to be transparent and provide comprehensive details about the data sources used, along with the collection, preprocessing, and preparation of the data for accurate data valuation. This will ensure enhanced comprehension and clarity in understanding how the data values were derived. Key subsections of this section include:

**Data information**   Details about how, where, when, and why the DVal candidate data was curated. This information includes details about the source and statistics of DVal data, its collection and curation process, licenses and privacy, and preprocessing. When applicable, the dataset datasheet Gebru et al. (2021) is included.

**Data preprocessing**   When preprocessing is applied to candidate data prior to data valuation, information is provided regarding the steps taken to prepare candidate data for valuation.

### H.4   DVAL METHOD

This section of the DValCard provides crucial information regarding the primary data valuation technique(s) and their usage. It contains a description of the method(s), including strengths, shortcomings, and characteristics e.g. runtime and space complexity, as well as the performance metric(s) used to evaluate the contribution of data points or groups of data points toward the desired quantification of data value. The performance metric function(s) may be model or data-driven. If applicable, it also contains a mathematical formulation of the method(s).

If the life cycle is model-driven, details are included pertaining to the learning algorithm(s), e.g., the model class, parameters, training procedure, and running time. If evaluation data is utilized by the data valuation technique(s), details are included in this section, including the evaluation data source, statistics, and preprocessing and cleaning procedures before data valuation. Key subsections of this section include:

**DVal technique(s)**   Provide information about the data valuation technique(s) with references, if appropriate.

**Performance metric(s)**   A description of the chosen performance metric(s) utilized to determine data value, with references when appropriate.

**Learning algorithm(s)**   A description of the learning algorithm(s) used in data valuation.

**Evaluation data**   Details about how, where, when, and why the evaluation data was curated.This information includes details about the source and statistics of evaluation data, its collection and curation process, licenses and privacy, and preprocessing. When applicable, the dataset datasheet Gebru et al. (2021) is included.

### H.5   DVAL REPORT

The DVal report includes a comprehensive analysis of both qualitative and quantitative aspects of raw or relative data values for a specific task or application. This analysis comprises the distributional analysis of data values, as well as an examination of how these values inform decisions related to the intended task - such as data removal or selection. The intended application for the data values significantly influences the output data values and which specific assessments are required. Key subsections of this section include:

**Data values**   Quantitative summary of data values, including the distribution of data values and their statistics, e.g., maximum/minimum data value.

**Removed/excluded instances**   Information regarding excluded data/instances, e.g., the diversity, distribution, and density of the excluded instances and discussions of how data values may have influenced those results. When applicable, data value distributions are reported by class and group/attribute (especially protected classes of individuals). Threshold values for exclusion are provided.

**Chosen/included instances**   Information regarding included data/instances, e.g., the diversity, distribution, and density of the included instances and discussions of how data values may have influenced those results. When applicable, data value distributions are reported by class and

group/attribute (especially protected classes of individuals). Threshold values for inclusion are provided.

## H.6 ETHICAL STATEMENT AND RECOMMENDATIONS

This section comprises ethical, legal, and environmental considerations, intended users, in- and out-of-scope use cases, and general recommendations for data valuation. Key subsections of this section include:

**Intended users and in/out-of-scope use cases**   Descriptions of the main stakeholders of the data valuation system and any (un)intended use cases of the resulted data values.

**Potential ethical issues to consider**   A concise discussion about the challenges and limitations of using the data values and potential impact on the intended task.

**Legal considerations**   At a minimum, details are included regarding permissions and licenses pertaining to the data valuation process.

**Environmental considerations**   A summary of the potential impact of the data valuation process on the environment, including details pertaining to GPU usage, when applicable.

**Recommendations**   A discussion of additional cautions intended users might consider as well as potential mitigation strategies.

