# OpenReview forum: "A case for data valuation transparency via DValCards"
_ICLR.cc/2025/Conference — Submitted to ICLR 2025_

### Official Review · Reviewer_Xakx · 2024-10-31

**Soundness:** 2
**Presentation:** 2
**Contribution:** 2
**Rating:** 3
**Confidence:** 4

**Summary:**

The paper studies data valuation and specifically the transparency of it. The authors highlight some issues of existing data valuation methods, in particular the bias of data values, which can result in technical and ethical consequences. The authors provide empirical evidence for such claims. The authors propose a framework called DValCards to encourage transparence in data valuation.

**Strengths:**

- The studied problem of data valuation is important and growing.
- The paper is relatively well written.
- The experimental results are with respect to real-world datasets.

**Weaknesses:**

- The claims against existing works are largely observational and empirical, and do not seem to be theoretically supported.
- The motivation for the DValCards can be made better. It seems that before Section 4, the authors are describing the issues with existing data valuation methods. In Section 4, where one might expect a mitigation or solution, the framework that does not seem to address these issues is described.
- Furthermore, the framework itself does not seem to be very extensively described or examined, in terms how it is applicable and beneficial.

**Questions:**

Is there a reason to limit to supervised classification? Is the method limited or more widely applicable to other settings?

In Section 3.1
`We find that varying the applied data imputation method results in appreciable variation of data val-
ues,`

And similar mentions of the instability of data valuation methods. The question is: Is the instability arising from the definition of the data valuation? Or from the estimation of the data valuation? The former suggests a fundamental methodological flaw of data valuation while the latter is due to the lack of better and more efficient computational techniques.

Following the previous question, if instability is a key limitation (of either data valuation, or estimation methods), specifically how does the proposed framework in Section 4 address it, by advocating for transparency?

---

### Official Review · Reviewer_7B5y · 2024-11-04

**Soundness:** 4
**Presentation:** 3
**Contribution:** 3
**Rating:** 5
**Confidence:** 3

**Summary:**

This paper investigates the properties of data valuation metrics, specifically their bias and instability, through case studies on real-world datasets. The authors highlight the limitations of data valuation metrics, including the impact of preprocessing techniques, minority groups, and technical and ethical side-effects. To address these limitations, they introduce DValCards, a standardized framework for reporting critical information and supporting decision-making about data valuation methods. The paper presents results on the instability of data valuation methods across different imputation techniques and highlights the implications of these inconsistencies using a case-study.

**Strengths:**

-   The authors provide a elaborate and comprehensive analysis of the impact of preprocessing techniques and class imbalance on data valuation metrics, especially imputation methods and their effects on class balance and rank stability. 12 Open-ML datasets are considered and 4 Data Valuation frameworks are chosen for comparison.
-   The introduction of DValCards is a valuable contribution to the field, providing a standardized framework for reporting critical information about data valuation metrics.
-   The paper raises important ethical considerations and implications of using data valuation metrics in context of a case study that highlights risks to undervalued groups.

**Weaknesses:**

-   The effectiveness of imputation preprocessing methods in standard data valuation tasks (eg. weighted training, noisy label detection) is not thoroughly evaluated, and the authors could provide more evidence. Instability of values is known in Data Valuation literature, but specifics with respect to imputation methods are not widely studied.
-   Since this paper is trying to unify a setting for all Data Valuation methods, it could benefit from expanding its scope to include runtime analysis (FLOPS analysis of the method), limitations with respect to scaling and tradeoff with performance. It would be worth including the impact of validation sets [1,2] on data value. It might be worth looking into other works to unify data valuation frameworks such as [3]
- The DVal Report in the DVal Card is reporting the data value range. However for a dataset, this may vary by just varying either the learning algorithm , or the performance metric or the valuation framework.  Data Values (especially their min max values) can have varying values but their rank stability, performance on standard data valuation tasks (noisy label detection or weighted training for instance) can help improve this part of the report.

[1] Kwon, Yongchan, and James Zou. "Data-oob: Out-of-bag estimate as a simple and efficient data value." International Conference on Machine Learning. PMLR, 2023.

[2] Jahagirdar, Himanshu, Jiachen T. Wang, and Ruoxi Jia. "Data Valuation in the Absence of a Reliable Validation Set." Transactions on Machine Learning Research.

[3] Jiang, Kevin, et al. "Opendataval: a unified benchmark for data valuation." Advances in Neural Information Processing Systems 36 (2023).

**Questions:**

-   Can the authors provide more information on whether imputation methods actually improve performance on standard valuation tasks ?
- It would be nice to use data valuation methods (used in certain places) instead of data valuation metrics (used more commonly in the paper), since they are generally referred to as frameworks.
- Can we see more examples of DVal Cards in this work? For a major contribution, the main paper has only one DVal Card and it seems to be a generic setting. It would be really interesting to see multiple DVal Cards and will reinforce the utility of having such a framework. A comparison

---

### Official Review · Reviewer_7aic · 2024-11-05

**Soundness:** 4
**Presentation:** 4
**Contribution:** 3
**Rating:** 6
**Confidence:** 4

**Summary:**

The paper conducts comprehensive empirical evaluations of existing data valuation metrics, identifying significant biases and instability in data-centric machine learning (ML). Key findings include: (1) common and inexpensive data pre-processing techniques can drastically change estimated data values; (2) subsampling using these metrics may exacerbate class imbalance; and (3) data valuation methods may undervalue data from underrepresented groups, raising ethical concerns. In particular, marginal contribution methods, such as Shapley-based approaches for tabular classification, demonstrate high variability due to data imputation preprocessing and may affect class balance and group fairness. To address these challenges and improve transparency, the paper introduces the novel Data Valuation Cards (DValCards).

**Strengths:**

The paper is well-motivated and conveys an essential message: existing data valuation methods, primarily designed for machine learning, may be unsuitable for data compensation in data markets. It highlights various practical challenges that emerge when these methods are repurposed for economic applications. Backed by comprehensive experimental analysis, the paper’s findings offer valuable insights and serve as practical guidelines for the effective design and implementation of data valuation metrics in data market contexts.

**Weaknesses:**

The paper raises an important issue, though its main limitation appears to be the lack of a fundamental solution. While DValCards help mitigate the issues of instability and fairness, they primarily serve as a more detailed documentation tool for data valuation methods.

The paper makes a valuable contribution by highlighting the challenges of existing data valuation approaches through extensive empirical evaluations, including issues related to instability, class imbalance, and fairness. However, some of these findings are not entirely unexpected. For instance, the instability of current metrics when different data imputations are applied is not very surprising: if the dataset changes, the data point values will change. In addition, it is not entirely clear why stability to data imputation should be considered an inherent property of a data valuation metric. Regarding fairness, it is not surprising that existing methods, which primarily aim to optimize test accuracy, might introduce bias. Nevertheless, the systematic evaluation using real-world data is valuable and provides an important, evidence-based perspective on these issues.

**Questions:**

None.

---

### Official Review · Reviewer_sEHS · 2024-11-06

**Soundness:** 2
**Presentation:** 2
**Contribution:** 2
**Rating:** 3
**Confidence:** 4

**Summary:**

This paper presents an empirical study of existing data valuation methods in terms of their sensitivity to pre-processing, the consequences of using them for data selection, and the tendency of undervaluing minorities.

**Strengths:**

- The paper is easy to follow.

**Weaknesses:**

- The paper's technical contribution is a bit limited, mainly focusing on evaluating existing methods.
- The findings from the paper are not novel. (1) Regarding the sensitivity to data imputation methods: data valuation fundamentally determines the contribution of a given data point based on the other data used together for training; hence, it is straightforward to see that the value of a data point would change depending on the choice of the imputation method because different imputation methods would change the formation of other data points. (2) Regarding the class imbalance: it is also natural that directly using data values to remove data would lead to class imbalance. This is because data valuation by design would assign same score to same data points. As a result, one would either remove two identical data points at the same time or keep them altogether, which in turn leads to a loss of balance in class representation. In fact, there has been existing work theoretically characterizes the limitation of using data valuation for data selection: https://arxiv.org/abs/2405.03875 (3) Regarding the last finding about undervaluing the minorities: this validity of this finding depends on the choice of validation data. If the validation comprises data points all from the underrepresented group, then the value of that group would be high instead of low as reported by the paper.

**Questions:**

See above.

---

### Meta-Review · Area_Chair_c4Bv · 2024-12-13

**Metareview:**

This paper studies the sensitivity of data valuation characteristics through an experimental study on classification datasets. The authors highlight how common approaches can lead to misinterpretation by drastically changing the values of these metrics, and proposes to reduce the potential for misuse by pairing datasets with "Data Valuation Cards."

## Strengths and Limitations:

- See below

## Recommendation

My recommendation is to accept the paper at this time given the feedback from reviewers and the lack of a rebuttal.

**Additional Comments On Reviewer Discussion:**

The paper did not receive a rebuttal from the authors, and there was no discussion with reviewers.

---

### Decision · Program_Chairs · 2025-01-22

Reject